# Eliciting User Preferences for Personalized Multi-Objective Decision Making through Comparative Feedback

**Han Shao**
TTIC
han@ttic.edu

**Lee Cohen**
TTIC
leecohencs@gmail.com

**Avrim Blum**
TTIC
avrim@ttic.edu

**Yishay Mansour**
Tel Aviv University and Google Research
mansour.yishay@gmail.com

**Aadirupa Saha**
TTIC
aadirupa@ttic.edu

**Matthew R. Walter**
TTIC
mwalter@ttic.edu

## Abstract

In this work, we propose a multi-objective decision making framework that accommodates different user preferences over objectives, where preferences are learned via policy comparisons. Our model consists of a known Markov decision process with a vector-valued reward function, with each user having an unknown preference vector that expresses the relative importance of each objective. The goal is to efficiently compute a near-optimal policy for a given user. We consider two user feedback models. We first address the case where a user is provided with two policies and returns their preferred policy as feedback. We then move to a different user feedback model, where a user is instead provided with two small weighted sets of representative trajectories and selects the preferred one. In both cases, we suggest an algorithm that finds a nearly optimal policy for the user using a number of comparison queries that scales quasilinearly in the number of objectives.

## 1 Introduction

Many real-world decision making problems involve optimizing over multiple objectives. For example, when designing an investment portfolio, one's investment strategy requires trading off maximizing expected gain with minimizing risk. When using Google Maps for navigation, people are concerned about various factors such as the worst and average estimated arrival time, traffic conditions, road surface conditions (e.g., whether or not it is paved), and the scenery along the way. As the advancement of technology gives rise to personalized machine learning (McAuley, 2022), in this paper, we design efficient algorithms for personalized multi-objective decision making.

While prior works have concentrated on approximating the Pareto-optimal solution set[1] (see Hayes et al. (2022) and Roijers et al. (2013) for surveys), we aim to find the optimal personalized policy for a user that reflects their unknown preferences over $k$ objectives. Since the preferences are unknown, we need to elicit users' preferences by requesting feedback on selected policies. The problem of eliciting preferences has been studied in Wang et al. (2022) using a strong query model that provides stochastic feedback on the quality of a single policy. In contrast, our work focuses on a more natural and intuitive query model, comparison queries, which query the user's preference over two selected policies, e.g., 'do you prefer a policy which minimizes average estimated arrival time or policy which minimizes the number of turns?'. Our goal is to find the optimal personalized policy using as few queries as possible.

---

[1]The Pareto-optimal solution set contains a optimal personalized policy for every possible user preference.

37th Conference on Neural Information Processing Systems (NeurIPS 2023).

To the best of our knowledge, we are the first to provide algorithms with theoretical guarantees for specific personalized multi-objective decision-making via policy comparisons.

Similar to prior works on multi-objective decision making, we model the problem using a finite-horizon Markov decision process (MDP) with a $k$-dimensional reward vector, where each entry is a non-negative scalar reward representative of one of the $k$ objectives. To account for user preferences, we assume that a user is characterized by a (hidden) $k$-dimensional *preference vector* with non-negative entries, and that the *personalized reward* of the user for each state-action is the inner product between this preference vector and the reward vector (for this state-action pair). We also distinguish between the $k$-dimensional value of a policy, which is the expected cumulative reward when selecting actions according to the policy, and the *personalized value* of a policy, which is the scalar expected cumulative personalized reward when selecting actions according to this policy. The MDP is known to the agent and the goal is to learn an optimal policy for the personalized reward function (henceforth, the *optimal personalized policy*) of a user via policy comparative feedback.

**Comparative feedback.** If people could clearly define their preferences over objectives (e.g., "my preference vector has 3 for the scenery objective, 2 for the traffic objective, and 1 for the road surface objective"), the problem would be easy—one would simply use the personalized reward function as a scalar reward function and solve for the corresponding policy. In particular, a similar problem with noisy feedback regarding the value of a single multi-objective policy (as mapping from states to actions) has been studied in Wang et al. (2022). As this type of fine-grained preference feedback is difficult for users to define, especially in environments where sequential decisions are made, we restrict the agent to rely solely on comparative feedback. Comparative feedback is widely used in practice. For example, ChatGPT asks users to compare two responses to improve its performance. This approach is more intuitive for users compared to asking for numerical scores of ChatGPT responses.

Indeed, as knowledge of the user's preference vector is sufficient to solve for their optimal personalized policy, the challenge is to learn a user's preference vector using a minimal number of easily interpretable queries. We therefore concentrate on comparison queries. The question is then what exactly to compare? Comparing state-action pairs might not be a good option for the aforementioned tasks—what is the meaning of two single steps in a route? Comparing single trajectories (e.g., routes in Google Maps) would not be ideal either. Consider for example two policies: one randomly generates either (personalized) GOOD or (personalized) BAD trajectories while the other consistently generates (personalized) MEDIOCRE trajectories. By solely comparing single trajectories without considering sets of trajectories, we cannot discern the user's preference regarding the two policies.

**Interpretable policy representation.** Since we are interested in learning preferences via policy comparison queries, we also suggest an alternative, more interpretable representation of a policy. Namely, we design an algorithm that given an explicit policy representation (as a mapping from states to distributions over actions), returns a weighted set of trajectories of size at most $k + 1$, such that its expected return is identical to the value of the policy.[2] It immediately follows from our formalization that for any user, the personalized return of the weighted trajectory set and the personalized value of the policy are also identical.

In this work, we focus on answering two questions:

*(1) How to find the optimal personalized policy by querying as few policy comparisons as possible?*
*(2) How can we find a more interpretable representation of policies efficiently?*

**Contributions.** In Section 2, we formalize the problem of eliciting user preferences and finding the optimal personalized policy via comparative feedback. As an alternative to an explicit policy representation, we propose a *weighted trajectory set* as a more interpretable representation. In Section 3, we provide an *efficient* algorithm for finding an approximate optimal personalized policy, where the policies are given by their formal representations, thus answering (1). In Section 4, we design two efficient algorithms that find the weighted trajectory set representation of a policy. Combined with the algorithm in Section 3, we have an algorithm for finding an approximate optimal personalized policy when policies are represented by weighted trajectory sets, thus answering (2).

**Related Work.** Multi-objective decision making has gained significant attention in recent years (see Roijers et al. (2013); Hayes et al. (2022) for surveys). Prior research has explored various approaches,

---

[2]A return of a trajectory is the cumulative reward obtained in the trajectory. The expectation in the expected return of a weighted trajectory set is over the weights.

such as assuming linear preferences or Bayesian Settings, or finding an approximated Pareto frontier. However, incorporating comparison feedback (as was done for Multi-arm bandits or active learning, e.g., Bengs et al. (2021) and Kane et al. (2017)) allows us a more comprehensive and systematic approach to handling different types of user preferences and provides (nearly) optimal personalized decision-making outcomes. We refer the reader to Appendix A for additional related work.

## 2   Problem Setup

**Sequential decision model.** We consider a Markov decision process (MDP) *known*[3] to the agent represented by a tuple $\langle \mathcal{S}, \mathcal{A}, s_0, P, R, H \rangle$, with finite state and action sets, $\mathcal{S}$ and $\mathcal{A}$, respectively, an initial state $s_0 \in \mathcal{S}$, and finite horizon $H \in \mathbb{N}$. For example, in the Google Maps example a state is an intersection and actions are turns. The transition function $P : \mathcal{S} \times \mathcal{A} \mapsto \text{Simplex}^{\mathcal{S}}$ maps state-action pairs into a state probability distribution. To model multiple objectives, the reward function $R : \mathcal{S} \times \mathcal{A} \mapsto [0,1]^k$ maps every state-action pair to a $k$-dimensional reward vector, where each component corresponds to one of the $k$ objectives (e.g., road surface condition, worst and average estimated arrival time). The *return* of a trajectory $\tau = (s_0, a_0, \ldots, s_{H-1}, a_{H-1}, s_H)$ is given by $\Phi(\tau) = \sum_{t=0}^{H-1} R(s_t, a_t)$.

A *policy* $\pi$ is a mapping from states to a distribution over actions. We denote the set of policies by $\Pi$. The *value* of a policy $\pi$, denoted by $V^\pi$, is the expected cumulative reward obtained by executing the policy $\pi$ starting from the initial state, $s_0$. Put differently, the value of $\pi$ is $V^\pi = V^\pi(s_0) = \mathbb{E}_{S_0=s_0} \left[ \sum_{t=0}^{H-1} R(S_t, \pi(S_t)) \right] \in [0, H]^k$, where $S_t$ is the random state at time step $t$ when executing $\pi$, and the expectation is over the randomness of $P$ and $\pi$. Note that $V^\pi = \mathbb{E}_\tau [\Phi(\tau)]$, where $\tau = (s_0, \pi(s_0), S_1, \ldots, \pi(S_{H-1}), S_H)$ is a random trajectory generated by executing $\pi$.

We assume the existence of a "do nothing" action $a_0 \in \mathcal{A}$, available only from the initial state $s_0$, that has zero reward for each objective $R(s_0, a_0) = \mathbf{0}$ and keeps the system in the initial state, i.e., $P(s_0 \mid s_0, a_0) = 1$ (e.g., this action corresponds to not commuting or refusing to play in a chess game.)[4]. We also define the (deterministic) "do nothing" policy $\pi_0$ that always selects action $a_0$ and has a value of $V^{\pi_0} = \mathbf{0}$. From a mathematical perspective, the assumption of "do nothing" ensures that $\mathbf{0}$ belongs to the value vector space $\{V^\pi | \pi \in \Pi\}$, which is precisely what we need.

Since the rewards are bounded between $[0, 1]$, we have that $1 \le \|V^\pi\|_2 \le \sqrt{k}H$ for every policy $\pi$. For convenience, we denote $C_V = \sqrt{k}H$. We denote by $d \le k$ the rank of the space spanned by all the value vectors obtained by $\Pi$, i.e., $d := \text{rank}(\text{span}(\{V^\pi | \pi \in \Pi\}))$.

**Linear preferences.** To incorporate personalized preferences over objectives, we assume each user is characterized by an unknown $k$-dimensional *preference vector* $w^* \in \mathbb{R}_+^k$ with a bounded norm $1 \le \|w^*\|_2 \le C_w$ for some (unknown) $C_w \ge 1$. We avoid assuming that $C_w = 1$ to accommodate for general linear rewards. Note that the magnitude of $w^*$ does not change the personalized optimal policy but affects the "indistinguishability" in the feedback model. By not normalizing $\|w^*\|_2 = 1$, we allow users to have varying levels of discernment. This preference vector encodes preferences over the multiple objectives and as a result, determines the user preferences over policies.

Formally, for a user characterized by $w^*$, the *personalized value* of policy $\pi$ is $\langle w^*, V^\pi \rangle \in \mathbb{R}_+$. We denote by $\pi^* := \arg\max_{\pi \in \Pi} \langle w^*, V^\pi \rangle$ and $v^* := \langle w^*, V^{\pi^*} \rangle$ the optimal *personalized policy* and its corresponding optimal personalized value for a user who is characterized by $w^*$. We remark that the "do nothing" policy $\pi_0$ (that always selects action $a_0$) has a value of $V^{\pi_0} = \mathbf{0}$, which implies a personalized value of $\langle w^*, V^{\pi_0} \rangle = 0$ for every $w^*$. For any two policies $\pi_1$ and $\pi_2$, the user characterized by $w^*$ prefers $\pi_1$ over $\pi_2$ if $\langle w^*, V^{\pi_1} - V^{\pi_2} \rangle > 0$. Our goal is to find the optimal personalized policy for a given user using as few interactions with them as possible.

**Comparative feedback.** Given two policies $\pi_1, \pi_2$, the user returns $\pi_1 \succ \pi_2$ whenever $\langle w^*, V^{\pi_1} - V^{\pi_2} \rangle > \epsilon$; otherwise, the user returns "indistinguishable" (i.e., whenever $|\langle w^*, V^{\pi_1} - V^{\pi_2} \rangle| \le \epsilon$). Here $\epsilon > 0$ measures the precision of the comparative feedback and

---

[3]If the MDP is unknown, one can approximate the MDP and apply the results of this work in the approximated MDP. More discussion is included in Appendix 5.

[4]We remark that the "do nothing" action require domain-specific knowledge, for example in the Google maps example the "do nothing" will be to stay put and in the ChatGPT example the "do nothing" is to answer nothing.

is small usually. The agent can query the user about their policy preferences using two different types of policy representations:

1. *Explicit policy representation of* $\pi$: An explicit representation of policy as mapping, $\pi : \mathcal{S} \to$ Simplex$^{\mathcal{A}}$.
2. *Weighted trajectory set representation of* $\pi$: A $\kappa$-sized set of trajectory-weight pairs $\{(p_i, \tau_i)\}_{i=1}^{\kappa}$ for some $\kappa \leq k + 1$ such that (i) the weights $p_1, \ldots, p_\kappa$ are non-negative and sum to 1; (ii) every trajectory in the set is in the support[5] of the policy $\pi$; and (iii) the expected return of these trajectories according to the weights is identical to the value of $\pi$, i.e., $V^\pi = \sum_{i=1}^{\kappa} p_i \Phi(\tau_i)$. Such comparison could be practical for humans. E.g., in the context of Google Maps, when the goal is to get from home to the airport, taking a specific route takes $40$ minutes $90\%$ of the time, but it can take 3 hours in case of an accident (which happens w.p. $10\%$) vs. taking the subway which always has a duration of 1 hour.

In both cases, the feedback is identical and depends on the hidden precision parameter $\epsilon$. As a result, the value of $\epsilon$ will affect the number of queries and how close the value of the resulting personalized policy is to the optimal personalized value. Alternatively, we can let the agent decide in advance on a maximal number of queries, which will affect the optimality of the returned policy.

**Technical Challenges.** In Section 3, we find an approximate optimal policy by $\mathcal{O}(\log \frac{1}{\epsilon})$ queries. To achieve this, we approach the problem in two steps. Firstly, we identify a set of linearly independent policy values, and then we estimate the preference vector $w^*$ using a linear program that incorporates comparative feedback. The estimation error of $w^*$ usually depends on the condition number of the linear program. Therefore, the main challenge we face is how to search for linear independent policy values that lead to a small condition number and providing a guarantee for this estimate.

In Section 4, we focus on how to design *efficient* algorithms to find the weighted trajectory set representation of a policy. Initially, we employ the well-known Carathéodory's theorem, which yields an inefficient algorithm with a potentially exponential running time of $|S|^H$. Our main challenge lies in developing an efficient algorithm with a running time of $\mathcal{O}(\text{poly}(H\,|S|\,|\mathcal{A}|))$. The approach based on Carathéodory's theorem treats the return of trajectories as independent $k$-dimensional vectors, neglecting the fact that they are all generated from the same MDP. To overcome this challenge, we leverage the inherent structure of MDPs.

## 3 Learning from Explicit Policies

In this section, we consider the case where the interaction with a user is based on explicit policy comparison queries. We design an algorithm that outputs a policy being nearly optimal for this user. For multiple different users, we only need to run part of the algorithm again and again. For brevity, we relegate all proofs to the appendix.

If the user's preference vector $w^*$ (up to a positive scaling) is given, then one can compute the optimal policy and its personalized value efficiently, e.g., using the Finite Horizon Value Iteration algorithm. In our work, $w^*$ is unknown and we interact with the user to learn $w^*$ through comparative feedback. Due to the structure model that is limited to meaningful feedback only when the compared policy values differ at least $\epsilon$, the exact value of $w^*$ cannot be recovered. We proceed by providing a high-level description of our ideas of how to estimate $w^*$.

(1) *Basis policies*: We find policies $\pi_1, \ldots, \pi_d$, and their respective values, $V^{\pi_1}, \ldots, V^{\pi_d} \in [0, H]^k$, such that their values are linearly independent and that together they span the entire space of value vectors.[6] These policies will not necessarily be personalized optimal for the current user, and instead serve only as building blocks to estimate the preference vector, $w^*$. In Section 3.1 we describe an algorithm that finds a set of basis policies for any given MDP.

(2) *Basis ratios*: For the basis policies, denote by $\alpha_i > 0$ the ratio between the personalized value of a *benchmark policy*, $\pi_1$, to the personalized value of $\pi_{i+1}$, i.e.,

$$\forall i \in [d-1] : \alpha_i \langle w^*, V^{\pi_1} \rangle = \langle w^*, V^{\pi_{i+1}} \rangle . \tag{1}$$

---

[5]We require the trajectories in this set to be in the support of the policy, to avoid trajectories that do not make sense, such as trajectories that "teleport" between different unconnected states (e.g., commuting at 3 mph in Manhattan in one state and then at 40 mph in New Orleans for the subsequent state).

[6]Recall that $d \leq k$ is the rank of the space spanned by all the value vectors obtained by all policies.

We will estimate $\widehat{\alpha}_i$ of $\alpha_i$ for all $i \in [d-1]$ using comparison queries. A detailed algorithm for estimating these ratios appears in Section 3.2. For intuition, if we obtain exact ratios $\widehat{\alpha}_i = \alpha_i$ for every $i \in [d-1]$, then we can compute the vector $\frac{w^*}{\|w^*\|_1}$ as follows. Consider the $d-1$ equations and $d-1$ variables in Eq (1). Since $d$ is the maximum number of value vectors that are linearly independent, and $V^{\pi_1}, \ldots V^{\pi_d}$ form a basis, adding the equation $\|w\|_1 = 1$ yields $d$ independent equations with $d$ variables, which allows us to solve for $w^*$. The details of computing an estimate of $w^*$ are described in Section 3.3.

## 3.1 Finding a Basis of Policies

In order to efficiently find $d$ policies with $d$ linearly independent value vectors that span the space of value vectors, one might think that selecting the $k$ policies that each optimizes one of the $k$ objectives will suffice. However, this might fail—in Appendix N, we show an instance in which these $k$ value vectors are linearly dependent even though there exist $k$ policies whose values span a space of rank $k$.

Moreover, our goal is to find not just any basis of policies, but a basis of policies such that (1) the personalized value of the benchmark policy $\langle w^*, V^{\pi_1} \rangle$ will be large (and hence the estimation error of ratio $\alpha_i$, $|\widehat{\alpha}_i - \alpha_i|$, will be small), and (2) that the linear program generated by this basis of policies and the basis ratios will produce a good estimate of $w^*$.

**Choice of $\pi_1$.** Besides linear independence of values, another challenge is to find a basis of policies to contain a benchmark policy, $\pi_1$ (where the index 1 is wlog) with a relatively large personalized value, $\langle w^*, V^{\pi_1} \rangle$, so that $\widehat{\alpha}_i$'s error is small (e.g., in the extreme case where $\langle w^*, V^{\pi_1} \rangle = 0$, we will not be able to estimate $\alpha_i$).

For any $w \in \mathbb{R}^k$, we use $\pi^w$ denote a policy that maximizes the scalar reward $\langle w, R \rangle$, i.e.,

$$\pi^w = \arg\max_{\pi \in \Pi} \langle w, V^\pi \rangle , \qquad (2)$$

and by $v^w = \langle w, V^{\pi^w} \rangle$ to denote the corresponding personalized value. Let $\mathbf{e}_1, \ldots, \mathbf{e}_k$ denote the standard basis. To find $\pi_1$ with large personalized value $\langle w^*, V^{\pi_1} \rangle$, we find policies $\pi^{\mathbf{e}_j}$ that maximize the $j$-th objective for every $j = 1, \ldots, k$ and then query the user to compare them until we find a $\pi^{\mathbf{e}_j}$ with (approximately) a maximal personalized value among them. This policy will be our benchmark policy, $\pi_1$. The details are described in lines 1–6 of Algorithm 1.

---

**Algorithm 1** Identification of Basis Policies

1: initialize $\pi^{\mathbf{e}^*} \leftarrow \pi^{\mathbf{e}_1}$
2: **for** $j = 2, \ldots, k$ **do**
3:   compare $\pi^{\mathbf{e}^*}$ and $\pi^{\mathbf{e}_j}$
4:   **if** $\pi^{\mathbf{e}_j} \succ \pi^{\mathbf{e}^*}$ **then** $\pi^{\mathbf{e}^*} \leftarrow \pi^{\mathbf{e}_j}$
5: **end for**
6: $\pi_1 \leftarrow \pi^{\mathbf{e}^*}$ and $u_1 \leftarrow \dfrac{V^{\pi^{\mathbf{e}^*}}}{\|V^{\pi^{\mathbf{e}^*}}\|_2}$
7: **for** $i = 2, \ldots, k$ **do**
8:   arbitrarily pick an orthonormal basis, $\rho_1, \ldots, \rho_{k+1-i}$, for $\text{span}(V^{\pi_1}, \ldots, V^{\pi_{i-1}})^\perp$.
9:   $j_{\max} \leftarrow \arg\max_{j \in [k+1-i]} \max(|v^{\rho_j}|, |v^{-\rho_j}|)$.
10:   **if** $\max(|v^{\rho_{j\max}}|, |v^{-\rho_{j\max}}|) > 0$ **then**
11:     $\pi_i \leftarrow \pi^{\rho_{j\max}}$ **if** $|v^{\rho_{j\max}}| > |v^{-\rho_{j\max}}|$; **otherwise** $\pi_i \leftarrow \pi^{-\rho_{j\max}}$. $u_i \leftarrow \rho_{j\max}$
12:   **else**
13:     **return** $(\pi_1, \pi_2, \ldots), (u_1, u_2, \ldots)$
14:   **end if**
15: **end for**

---

**Choice of $\pi_2, \ldots, \pi_d$.** After finding $\pi_1$, we next search the remaining $d-1$ polices $\pi_2, \ldots, \pi_d$ sequentially (lines 8–13 of Algorithm 1). For $i = 2, \ldots, d$, we find a direction $u_i$ such that (i) the vector $u_i$ is orthogonal to the space of current value vectors $\text{span}(V^{\pi_1}, \ldots, V^{\pi_{i-1}})$, and (ii) there exists a policy $\pi_i$ such that $V^{\pi_i}$ has a significant component in the direction of $u_i$. Condition (i) is used to guarantee that the policy $\pi_i$ has a value vector linearly independent of $\text{span}(V^{\pi_1}, \ldots, V^{\pi_{i-1}})$. Condition (ii) is used to cope with the error caused by inaccurate approximation of the ratios $\widehat{\alpha}_i$. Intuitively, when $\|\alpha_i V^{\pi_1} - V^{\pi_{i+1}}\|_2 \ll \epsilon$, the angle between $\widehat{\alpha}_i V^{\pi_1} - V^{\pi_{i+1}}$ and $\alpha_i V^{\pi_1} - V^{\pi_{i+1}}$ could be very large, which results in an inaccurate estimate of $w^*$ in the direction of $\alpha_i V^{\pi_1} - V^{\pi_{i+1}}$. For example, if $V^{\pi_1} = \mathbf{e}_1$ and $V^{\pi_i} = \mathbf{e}_1 + \frac{1}{w_i^*}\epsilon \mathbf{e}_i$ for $i = 2, \ldots, k$, then $\pi_1, \pi_i$ are "indistinguishable" and the estimate ratio $\widehat{\alpha}_{i-1}$ can be 1. Then the estimate of $w^*$ by solving linear equations in Eq (1) is $(1, 0, \ldots, 0)$, which could be far from the true $w^*$. Finding $u_i$'s in which policy values have a large component can help with this problem.

Algorithm 1 provides a more detailed description of this procedure. Note that if there are $n$ different users, we will run Algorithm 1 at most $k$ times instead of $n$ times. The reason is that Algorithm

1 only utilizes preference comparisons while searching for $\pi_1$ (lines 1-6), and not for $\pi_2, \ldots, \pi_k$ (which contributes to the $k^2$ factor in computational complexity). As there are at most $k$ candidates for $\pi_1$, namely $\pi^{e_1}, \ldots, \pi^{e_k}$, we execute lines 1-6 of Algorithm 1 for $n$ rounds and lines 8-13 for only $k$ rounds.

## 3.2 Computation of Basis Ratios

As we have mentioned before, comparing basis policies alone does not allow for the exact computation of the $\alpha_i$ ratios as comparing $\pi_1, \pi_i$ can only reveal which is better but not how much. To this end, we will use the "do nothing" policy to approximate every ratio $\alpha_i$ up to some additive error $|\widehat{\alpha}_i - \alpha_i|$ using binary search over the parameter $\widehat{\alpha}_i \in [0, C_\alpha]$ for some $C_\alpha \geq 1$ (to be determined later) and comparison queries of policy $\pi_{i+1}$ with policy $\widehat{\alpha}_i \pi_1 + (1 - \widehat{\alpha}_i)\pi_0$ if $\widehat{\alpha}_i \leq 1$ (or comparing $\pi_1$ and $\frac{1}{\widehat{\alpha}_i}\pi_{i+1} + (1 - \frac{1}{\widehat{\alpha}_i})\pi_0$ instead if $\widehat{\alpha}_i > 1$).[7] Notice that the personalized value of $\widehat{\alpha}_i \pi_1 + (1 - \widehat{\alpha}_i)\pi_0$ is identical to the personalized value of $\pi_1$ multiplied by $\widehat{\alpha}_i$. We stop once $\widehat{\alpha}_i$ is such that the user returns "indistinguishable". Once we stop, the two policies have roughly the same personalized value,

$$\text{if } \widehat{\alpha}_i \leq 1, |\widehat{\alpha}_i \langle w^*, V^{\pi_1} \rangle - \langle w^*, V^{\pi_{i+1}} \rangle| \leq \epsilon \, ; \quad \text{if } \widehat{\alpha}_i > 1, \left| \langle w^*, V^{\pi_1} \rangle - \frac{1}{\widehat{\alpha}_i} \langle w^*, V^{\pi_{i+1}} \rangle \right| \leq \epsilon. \quad (3)$$

Eq (1) combined with the above inequality implies that $|\widehat{\alpha}_i - \alpha_i| \langle w^*, V^{\pi_1} \rangle \leq C_\alpha \epsilon$. Thus, the approximation error of each ratio is bounded by $|\widehat{\alpha}_i - \alpha_i| \leq \frac{C_\alpha \epsilon}{\langle w^*, V^{\pi_1} \rangle}$. To make sure the procedure will terminate, we need to set $C_\alpha \geq \frac{v^*}{\langle w^*, V^{\pi_1} \rangle}$ since $\alpha_i$'s must lie in the interval $[0, \frac{v^*}{\langle w^*, V^{\pi_1} \rangle}]$. Upon stopping binary search once Eq (3) holds, it takes at most $\mathcal{O}(d \log(C_\alpha \langle w^*, V^{\pi_1} \rangle / \epsilon))$ comparison queries to estimate all the $\alpha_i$'s.

Due to the carefully picked $\pi_1$ in Algorithm 1, we can upper bound $\frac{v^*}{\langle w^*, V^{\pi_1} \rangle}$ by $2k$ and derive an upper bound for $|\widehat{\alpha}_i - \alpha_i|$ by selecting $C_\alpha = 2k$.

**Lemma 1.** *When $\epsilon \leq \frac{v^*}{2k}$, we have $\frac{v^*}{\langle w^*, V^{\pi_1} \rangle} \leq 2k$, and $|\widehat{\alpha}_i - \alpha_i| \leq \frac{4k^2 \epsilon}{v^*}$ for every $i \in [d]$.*

In what follows we set $C_\alpha = 2k$. The pseudo code of the above process of estimating $\alpha$'s is deferred to Algorithm 3 in Appendix C.

## 3.3 Preference Approximation and Personalized Policy

We move on to present an algorithm that estimates $w^*$ and calculates a nearly optimal personalized policy. Given the $\pi_i$'s returned by Algorithm 1 and the $\widehat{\alpha}_i$'s returned by Algorithm 3, consider a matrix $\widehat{A} \in \mathbb{R}^{d \times k}$ with 1st row $V^{\pi_1 \top}$ and the $i$th row $(\widehat{\alpha}_{i-1} V^{\pi_1} - V^{\pi_i})^\top$ for every $i = 2, \ldots, d$. Let $\widehat{w}$ be a solution to $\widehat{A}x = \mathbf{e}_1$. We will show that $\widehat{w}$ is a good estimate of $w' := \frac{w^*}{\langle w^*, V^{\pi_1} \rangle}$ and that $\pi^{\widehat{w}}$ is a nearly optimal personalized policy. In particular, when $\epsilon$ is small, we have $|\langle \widehat{w}, V^\pi \rangle - \langle w', V^\pi \rangle| = \mathcal{O}(\epsilon^{\frac{1}{3}})$ for every policy $\pi$. Putting this together, we derive the following theorem.

**Theorem 1.** *Consider the algorithm of computing $\widehat{A}$ and any solution $\widehat{w}$ to $\widehat{A}x = \mathbf{e}_1$ and outputting the policy $\pi^{\widehat{w}} = \arg\max_{\pi \in \Pi} \langle \widehat{w}, V^\pi \rangle$, which is the optimal personalized policy for preference vector $\widehat{w}$. Then the output policy $\pi^{\widehat{w}}$ satisfying that $v^* - \left\langle w^*, V^{\pi^{\widehat{w}}} \right\rangle \leq \mathcal{O}\left( \left(\sqrt{k} + 1\right)^{d + \frac{14}{3}} \epsilon^{\frac{1}{3}} \right)$ using $\mathcal{O}(k \log(k/\epsilon))$ comparison queries.*

**Computation Complexity** We remark that Algorithm 1 solves Eq (2) for the optimal policy in scalar reward MDP at most $\mathcal{O}(k^2)$ times. Using, e.g., Finite Horizon Value iteration to solve for the optimal policy takes $\mathcal{O}(H|\mathcal{S}|^2|\mathcal{A}|)$ steps. However, while the time complexity it takes to return the optimal policy for a single user is $\mathcal{O}(k^2 H|\mathcal{S}|^2|\mathcal{A}| + k \log(\frac{k}{\epsilon}))$, considering $n$ different users rather than one results in overall time complexity of $\mathcal{O}((k^3 + n)H|\mathcal{S}|^2|\mathcal{A}| + nk \log(\frac{k}{\epsilon}))$.

**Proof Technique.** The standard technique typically starts by deriving an upper bound on $\|\widehat{w} - w^*\|_2$ and then uses this bound to upper bound $\sup_\pi |\langle \widehat{w}, V^\pi \rangle - \langle w^*, V^\pi \rangle|$ as $C_V \|\widehat{w} - w^*\|_2$. However,

---

[7]We write $\widehat{\alpha}_i \pi_1 + (1 - \widehat{\alpha}_i)\pi_0$ to indicate that $\pi_1$ is used with probability $\widehat{\alpha}_i$, and that $\pi_0$ is used with probability $1 - \widehat{\alpha}_i$.

this method fails to achieve a non-vacuous bound in case there are two basis policies that are close to each other. For instance, consider the returned basis policy values: $V^{\pi_1} = (1,0,0)$, $V^{\pi_2} = (1,1,0)$, and $V^{\pi_3} = (1,0,\eta)$ for some $\eta > 0$. When $\eta$ is extremely small, the estimate $\widehat{w}$ becomes highly inaccurate in the direction of $(0,0,1)$, leading to a large $\|\widehat{w} - w^*\|_2$. Even in such cases, we can still obtain a non-vacuous guarantee since the selection of $V^{\pi_3}$ (line 11 of Algorithm 1) implies that no policy in $\Pi$ exhibits a larger component in the direction of $(0,0,1)$ than $\pi_3$.

**Proof Sketch.** The analysis of Theorem 1 has two parts. First, as mentioned in Sec 3.1, when $\|\alpha_i V^{\pi_1} - V^{\pi_{i+1}}\|_2 \ll \epsilon$, the error of $\widehat{\alpha}_{i+1}$ can lead to inaccurate estimate of $w^*$ in direction $\alpha_i V^{\pi_1} - V^{\pi_{i+1}}$. Thus, we consider another estimate of $w^*$ based only on some $\pi_{i+1}$'s with a relatively large $\|\alpha_i V^{\pi_1} - V^{\pi_{i+1}}\|_2$. In particular, for any $\delta > 0$, let $d_\delta := \min_{i \geq 2:\max(|v^{u_i}|, |v^{-u_i}|) \leq \delta} i - 1$. That is to say, for $i = 2, \ldots, d_\delta$, the policy $\pi_i$ satisfies $\langle u_i, V^{\pi_i} \rangle > \delta$ and for any policy $\pi$, we have $\langle u_{d_\delta+1}, V^\pi \rangle \leq \delta$. Then, for any policy $\pi$ and any unit vector $\xi \in \mathrm{span}(V^{\pi_1}, \ldots, V^{\pi_{d_\delta}})^\perp$, we have $\langle \xi, V^\pi \rangle \leq \sqrt{k}\delta$. This is because at round $d_\delta + 1$, we pick an orthonormal basis $\rho_1, \ldots, \rho_{k-d_\delta}$ of $\mathrm{span}(V^{\pi_1}, \ldots, V^{\pi_{d_\delta}})^\perp$ (line 8 in Algorithm 1) and pick $u_{d_\delta+1}$ to be the one in which there exists a policy with the largest component as described in line 9. Hence, $|\langle \rho_j, V^\pi \rangle| \leq \delta$ for all $j \in [k - d_\delta]$. Then, we have $\langle \xi, V^\pi \rangle = \sum_{j=1}^{k-d_\delta} \langle \xi, \rho_j \rangle \langle \rho_j, V^\pi \rangle \leq \sqrt{k}\delta$ by Cauchy-Schwarz inequality. Let $\widehat{A}^{(\delta)} \in \mathbb{R}^{d_\delta \times k}$ be the sub-matrix comprised of the first $d_\delta$ rows of $\widehat{A}$. Then we consider an alternative estimate $\widehat{w}^{(\delta)} = \arg\min_{x:\widehat{A}^{(\delta)}x=\mathbf{e}_1} \|x\|_2$, the minimum norm solution of $x$ to $\widehat{A}^{(\delta)}x = \mathbf{e}_1$. We upper bound $\sup_\pi |\langle \widehat{w}^{(\delta)}, V^\pi \rangle - \langle w', V^\pi \rangle|$ in Lemma 2 and $\sup_\pi |\langle \widehat{w}, V^\pi \rangle - \langle \widehat{w}^{(\delta)}, V^\pi \rangle|$ in Lemma 3. Then we are done with the proof of Theorem 1.

**Lemma 2.** *If $|\widehat{\alpha}_i - \alpha_i| \leq \epsilon_\alpha$ and $\alpha_i \leq C_\alpha$ for all $i \in [d-1]$, for every $\delta \geq 4C_\alpha^{\frac{2}{3}} C_V d^{\frac{1}{3}} \epsilon_\alpha^{\frac{1}{3}}$, we have*

$$\left|\langle \widehat{w}^{(\delta)}, V^\pi \rangle - \langle w', V^\pi \rangle\right| \leq \mathcal{O}\left(\frac{C_\alpha C_V^4 d^{\frac{3}{2}}_\delta \|w'\|_2^2 \epsilon_\alpha}{\delta^2} + \sqrt{k}\delta \|w'\|_2\right) \text{ for all } \pi, \text{ where } w' = \frac{w^*}{\langle w^*, V^{\pi_1} \rangle}.$$

Since we only remove the rows in $\widehat{A}$ corresponding to $u_i$'s in the subspace where no policy's value has a large component, $\widehat{w}$ and $\widehat{w}^{(\delta)}$ are close in terms of $\sup_\pi |\langle \widehat{w}, V^\pi \rangle - \langle \widehat{w}^{(\delta)}, V^\pi \rangle|$.

**Lemma 3.** *If $|\widehat{\alpha}_i - \alpha_i| \leq \epsilon_\alpha$ and $\alpha_i \leq C_\alpha$ for all $i \in [d-1]$, for every policy $\pi$ and every $\delta \geq 4C_\alpha^{\frac{2}{3}} C_V d^{\frac{1}{3}} \epsilon_\alpha^{\frac{1}{3}}$, we have $|\widehat{w} \cdot V^\pi - \widehat{w}^{(\delta)} \cdot V^\pi| \leq \mathcal{O}((\sqrt{k}+1)^{d-d_\delta} C_\alpha \epsilon^{(\delta)})$, where $\epsilon^{(\delta)} = \frac{C_\alpha C_V^4 d^{\frac{3}{2}}_\delta \|w'\|_2^2 \epsilon_\alpha}{\delta^2} + \sqrt{k}\delta \|w'\|_2$ is the upper bound in Lemma 2.*

Note that the result in Theorem 1 has a factor of $k^{\frac{d}{2}}$, which is exponential in $d$. Usually, we consider the case where $k = \mathcal{O}(1)$ is small and thus $k^d = \mathcal{O}(1)$ is small. We get rid of the exponential dependence on $d$ by applying $\widehat{w}^{(\delta)}$ to estimate $w^*$ directly, which requires us to set the value of $\delta$ beforehand. The following theorem follows directly by assigning the optimal value for $\delta$ in Lemma 2.

**Theorem 2.** *Consider the algorithm of computing $\widehat{A}$ and any solution $\widehat{w}^{(\delta)}$ to $\widehat{A}^{(\delta)}x = \mathbf{e}_1$ for $\delta = k^{\frac{5}{3}}\epsilon^{\frac{1}{3}}$ and outputting the policy $\pi^{\widehat{w}^{(\delta)}} = \arg\max_{\pi \in \Pi} \langle \widehat{w}^{(\delta)}, V^\pi \rangle$. Then the policy $\pi^{\widehat{w}^{(\delta)}}$ satisfies that $v^* - \langle w^*, V^{\pi^{\widehat{w}^{(\delta)}}} \rangle \leq \mathcal{O}\left(k^{\frac{13}{6}}\epsilon^{\frac{1}{3}}\right)$.*

Notice that the algorithm in Theorem 2 needs to set the hyperparameter $\delta$ beforehand while we don't have to set any hyperparameter in Theorem 1. The improper value of $\delta$ could degrade the performance of the algorithm. But we can approximately estimate $\epsilon$ by binary searching $\eta \in [0,1]$ and comparing $\pi_1$ against the scaled version of itself $(1-\eta)\pi_1$ until we find an $\eta$ such that the user cannot distinguish between $\pi_1$ and $(1-\eta)\pi_1$. Then we can obtain an estimate of $\epsilon$ and use the estimate to set the hyperparameter.

We remark that though we think of $k$ as a small number, it is unclear whether the dependency on $\epsilon$ in Theorems 1 and 2 is optimal. The tight dependency on $\epsilon$ is left as an open problem. We briefly discuss a potential direction to improve this bound in Appendix I.

## 4  Learning from Weighted Trajectory Set Representation

In the last section, we represented policies using their explicit form as state-action mappings. However, such a representation could be challenging for users to interpret. For example, how safe is a car

described by a list of $|\mathcal{S}|$ states and actions such as "turning left"? In this section, we design algorithms that return a more interpretable policy representation—a weighted trajectory set.

Recall the definition in Section 2, a weighted trajectory set is a small set of trajectories from the support of the policy and corresponding weights, with the property that the expected return of the trajectories in the set (according to the weights) is **exactly** the value of the policy (henceforth, the *exact value property*).[8] As these sets preserve all the information regarding the multi-objective values of policies, they can be used as policy representations in policy comparison queries of Algorithm 3 without compromising on feedback quality. Thus, using these representations obtain the same optimality guarantees regarding the returned policy in Section 3 (but would require extra computation time to calculate the sets).

There are two key observations on which the algorithms in this section are based:

(1) Each policy $\pi$ induces a distribution over trajectories. Let $q^\pi(\tau)$ denote the probability that a trajectory $\tau$ is sampled when selecting actions according to $\pi$. The expected return of all trajectories under $q^\pi$ is identical to the value of the policy, i.e., $V^\pi = \sum_\tau q^\pi(\tau)\Phi(\tau)$. In particular, the value of a policy is a convex combination of the returns of the trajectories in its support. However, we avoid using this convex combination to represent a policy since the number of trajectories in the support of a policy could be exponential in the number of states and actions.

(2) The existence of a small weighted trajectory set is implied by Carathéodory's theorem. Namely, since the value of a policy is in particular a convex combination of the returns of the trajectories in its support, Carathéodory's theorem implies that there exist $k+1$ trajectories in the support of the policy and weights for them such that a convex combination of their returns is the value of the policy. Such a $(k+1)$-sized set will be the output of our algorithms.

We can apply the idea behind Carathéodory's theorem proof to compress trajectories as follows. For any $(k+2)$-sized set of $k$-dimensional vectors $\{\mu_1, \ldots, \mu_{k+2}\}$, for any convex combination of them $\mu = \sum_{i=1}^{k+2} p_i\mu_i$, we can always find a $(k+1)$-sized subset such that $\mu$ can be represented as the convex combination of the subset by solving a linear equation. Given an input of a probability distribution $p$ over a set of $k$-dimensional vectors, $M$, we pick $k+2$ vectors from $M$, reduce at least one of them through the above procedure. We repeat this step until we are left with at most $k+1$ vectors. We refer to this algorithm as C4 (Compress Convex Combination using Carathéodory's theorem). The pseudocode is described in Algorithm 4, which is deferred to Appendix J due to space limit. The construction of the algorithm implies the following lemma immediately.

**Lemma 4.** *Given a set of $k$-dimensional vectors $M \subset \mathbb{R}^k$ and a distribution $p$ over $M$, $C4(M, p)$ outputs $M' \subset M$ with $|M'| \leq k+1$ and a distribution $q \in \mathrm{Simplex}^{M'}$ satisfying that $\mathbb{E}_{\mu \sim q}[\mu] = \mathbb{E}_{\mu \sim p}[\mu]$ in time $\mathcal{O}(|M|\,k^3)$.*

So now we know how to compress a set of trajectories to the desired size. The main challenge is how to do it **efficiently** (in time $\mathcal{O}(\mathrm{poly}(H\,|S|\,|\mathcal{A}|))$). Namely, since the set of all trajectory returns from the support of the policy could be of size $\Omega(|\mathcal{S}|^H)$, using it as input to C4 Algorithm is inefficient. Instead, we will only use C4 as a subroutine when the number of trajectories is small.

We propose two efficient approaches for finding weighted trajectory representations. Both approaches take advantage of the property that all trajectories are generated from the same policy on the same MDP. First, we start with a small set of trajectories of length of 1, expand them, compress them, and repeat until we get the set of trajectory lengths of $H$. The other is based on the construction of a layer graph where a policy corresponds to a flow in this graph and we show that finding representative trajectories is equivalent to flow decomposition.

In the next subsection, we will describe the expanding and compressing approach and defer the flow decomposition based approach to Appendix K due to space considerations. We remark that the flow decomposition approach has a running time of $\mathcal{O}(H^2\,|\mathcal{S}|^2 + k^3H\,|\mathcal{S}|^2)$ (see appendix for

---

details), which underperforms the expanding and compressing approach (see Theorem 3) whenever $|\mathcal{S}|H + |\mathcal{S}|k^3 = \omega(k^4 + k|\mathcal{S}|)$. For presentation purposes, in the following, we only consider the deterministic policies. Our techniques can be easily extended to random policies.[9]

[8] Without asking for the exact value property, one could simply return a sample of $\mathcal{O}(\log k/(\epsilon')^2)$ trajectories from the policy and uniform weights. With high probability, the expected return of every objective is $\epsilon'$-far from its expected value. The problem is that this set does not necessarily capture rare events. For example, if the probability of a crash for any car is between $(\epsilon')^4$ and $(\epsilon')^2$, depending on the policy, users that only care about safety (i.e., no crashes) are unlikely to observe any "unsafe" trajectories at all, in which case we would miss valuable feedback.

**Expanding and Compressing Approach.**

The basic idea is to find $k + 1$ trajectories of length 1 to represent $V^\pi$ first and then increase the length of the trajectories without increasing the number of trajectories. For policy $\pi$, let $V^\pi(s, h) = \mathbb{E}_{S_0=s}\left[\sum_{t=0}^{h-1} R(S_t, \pi(S_t))\right]$ be the value of $\pi$ with initial state $S_0 = s$ and time horizon $h$. Since we study the representation for a fixed policy $\pi$ in this section, we slightly abuse the notation and represent a trajectory by $\tau^\pi = (s, s_1, \ldots, s_H)$. We denote the state of trajectory $\tau$ at time $t$ as $s_t^\tau = s_t$. For a trajectory prefix $\tau = (s, s_1, \ldots, s_h)$ of $\pi$ with initial state $s$ and $h \le H$ subsequent states, the return of $\tau$ is $\Phi(\tau) = R(s, \pi(s)) + \sum_{t=1}^{h-1} R(s_t, \pi(s_t))$. Let $J(\tau)$ be the expected return of trajectories (of length $H$) with the prefix being $\tau$, i.e.,

$$J(\tau) := \Phi(\tau) + V(s_h^\tau, H - h).$$

For any $s \in \mathcal{S}$, let $\tau \circ s$ denote the trajectory of appending $s$ to $\tau$. We can solve $V^\pi(s, h)$ for all $s \in \mathcal{S}, h \in [H]$ by dynamic programming in time $\mathcal{O}(kH|\mathcal{S}|^2)$. Specifically, according to definition, we have $V^\pi(s, 1) = R(s, \pi(s))$ and

$$V^\pi(s, h + 1) = R(s, \pi(s)) + \sum_{s' \in \mathcal{S}} P(s'|s, \pi(s))V^\pi(s', h). \tag{4}$$

Thus, we can represent $V^\pi$ by

$$V^\pi = R(s_0, \pi(s_0)) + \sum_{s \in \mathcal{S}} P(s|s_0, \pi(s_0))V^\pi(s, H - 1) = \sum_{s \in \mathcal{S}} P(s|s_0, \pi(s_0))J(s_0, s).$$

By applying C4, we can find a set of representative trajectories of length 1, $F^{(1)} \subset \{(s_0, s)|s \in \mathcal{S}\}$, with $|F^{(1)}| \le k + 1$ and weights $\beta^{(1)} \in \mathrm{Simplex}^{F^{(1)}}$ such that

$$V^\pi = \sum_{\tau \in F^{(1)}} \beta^{(1)}(\tau)J(\tau). \tag{5}$$

Supposing that we are given a set of trajectories $F^{(t)}$ of length $t$ with weights $\beta^{(t)}$ such that $V^\pi = \sum_{\tau \in F^{(t)}} \beta^{(t)}(\tau)J(\tau)$, we can first increase the length of trajectories by 1 through Eq (4) and obtain a subset of $\{\tau \circ s | \tau \in F^{(t)}, s \in \mathcal{S}\}$, in which the trajectories are of length $t + 1$. Specifically, we have

$$V^\pi = \sum_{\tau \in F^{(t)}, s \in \mathcal{S}} \beta^{(t)}(\tau)P(s|s_t^\tau, \pi(s_t^\tau))J(\tau \circ s). \tag{6}$$

Then we would like to compress the above convex combination through C4 as we want to keep track of at most $k + 1$ trajectories of length $t + 1$ due to the computing time. More formally, let $J_{F^{(t)}} := \{J(\tau \circ s)|\tau \in F^{(t)}, s \in \mathcal{S}\}$ be the set of expected returns and $p_{F^{(t)}, \beta^{(t)}} \in \mathrm{Simplex}^{F^{(t)} \times \mathcal{S}}$ with $p_{F^{(t)}, \beta^{(t)}}(\tau \circ s) = \beta^{(t)}(\tau)P(s|s_t^\tau, \pi(s_t^\tau))$ be the weights appearing in Eq (6). Here $p_{F^{(t)}, \beta^{(t)}}$ defines a distribution over $J_{F^{(t)}}$ with the probability of drawing $J(\tau \circ s)$ being $p_{F^{(t)}, \beta^{(t)}}(\tau \circ s)$. Then we can apply C4 over $(J_{F^{(t)}}, p_{F^{(t)}, \beta^{(t)}})$ and compress the representative trajectories $\{\tau \circ s | \tau \in F^{(t)}, s \in \mathcal{S}\}$. We start with trajectories of length 1 and repeat the process of expanding and compressing until we get trajectories of length $H$. The details are described in Algorithm 2.

---

[9]In Section 3, we consider a special type of random policy that is a mixed strategy of a deterministic policy (the output from an algorithm that solves for the optimal policy for an MDP with scalar reward) with the "do nothing" policy. For this specific random policy, we can find weighted trajectory representations for both policies and then apply Algorithm 4 to compress the representation.

---

**Algorithm 2** Expanding and compressing trajectories

---
1: compute $V^\pi(s, h)$ for all $s \in \mathcal{S}, h \in [H]$ by dynamic programming according to Eq (4)
2: $F^{(0)} = \{(s_0)\}$ and $\beta^{(0)}(s_0) = 1$
3: **for** $t = 0, \ldots, H-1$ **do**
4:  $J_{F^{(t)}} \leftarrow \{J(\tau \circ s) | \tau \in F^{(t)}, s \in \mathcal{S}\}$ and $p_{F^{(t)}, \beta^{(t)}}(\tau \circ s) \leftarrow \beta^{(t)}(\tau) P(s | s_t^\tau, \pi(s_t^\tau))$ for
   $\tau \in F^{(t)}, s \in \mathcal{S}$                 // expanding step
5:  $(J^{(t+1)}, \beta^{(t+1)}) \leftarrow \text{C4}(J_{F^{(t)}}, p_{F^{(t)}, \beta^{(t)}})$ and $F^{(t+1)} \leftarrow \{\tau | J(\tau) \in J^{(t+1)}\}$  // compressing
   step
6: **end for**
7: output $F^{(H)}$ and $\beta^{(H)}$

---

**Theorem 3.** *Algorithm 2 outputs $F^{(H)}$ and $\beta^{(H)}$ satisfying that $\left|F^{(H)}\right| \leq k+1$ and $\sum_{\tau \in F^{(H)}} \beta^{(H)}(\tau) \Phi(\tau) = V^\pi$ in time $\mathcal{O}(k^4 H |\mathcal{S}| + kH |\mathcal{S}|^2)$.*

The proof of Theorem 3 follows immediately from the construction of the algorithm. According to Eq (5), we have $V^\pi = \sum_{\tau \in F^{(1)}} \beta^{(1)}(\tau) J(\tau)$. Then we can show that the output of Algorithm 2 is a valid weighted trajectory set by induction on the length of representative trajectories. C4 guarantees that $\left|F^{(t)}\right| \leq k+1$ for all $t = 1, \ldots, H$, and thus, we only keep track of at most $k+1$ trajectories at each step and achieve the computation guarantee in the theorem. Combined with Theorem 1, we derive the following Corollary.

**Corollary 1.** *Running the algorithm in Theorem 1 with weighted trajectory set representation returned by Algorithm 2 gives us the same guarantee as that of Theorem 1 in time $\mathcal{O}(k^2 H |\mathcal{S}|^2 |\mathcal{A}| + (k^5 H |\mathcal{S}| + k^2 H |\mathcal{S}|^2) \log(\frac{k}{\epsilon}))$.*

## 5 Discussion

In this paper, we designed efficient algorithms for learning users' preferences over multiple objectives from comparative feedback. The efficiency is expressed in both the running time and number of queries (both polynomial in $H, |\mathcal{S}|, |\mathcal{A}|, k$ and logarithmic in $1/\epsilon$). The learned preferences of a user can then be used to reduce the problem of finding a personalized optimal policy for this user to a (finite horizon) single scalar reward MDP, a problem with a known efficient solution. As we have focused on minimizing the policy comparison queries, our algorithms are based on polynomial time pre-processing calculations that save valuable comparison time for users.

The results in Section 3 are of independent interest and can be applied to a more general learning setting, where for some unknown linear parameter $w^*$, given a set of points $X$ and access to comparison queries of any two points, the goal is to learn $\arg\max_{x \in X} \langle w^*, x \rangle$. E.g., in personalized recommendations for coffee beans in terms of the coffee profile described by the coffee suppliers (body, aroma, crema, roast level,...), while users could fail to describe their optimal coffee beans profile, adopting the methodology in Section 3 can retrieve the ideal coffee beans for a user using comparisons (where the mixing with "do nothing" is done by diluting the coffee with water and the optimal coffee for a given profile is the one closest to it).

When moving from the explicit representation of policies as mappings from states to actions to a more natural policy representation as a weighted trajectory set, we then obtained the same optimality guarantees in terms of the number of queries. While there could be other forms of policy representations (e.g., a small subset of common states), one advantage of our weighted trajectory set representation is that it captures the essence of the policy multi-objective value in a clear manner via $\mathcal{O}(k)$ trajectories and weights. The algorithms provided in Section 4 are standalone and could also be of independent interest for explainable RL (Alharin et al., 2020). For example, to exemplify the multi-objective performance of generic robotic vacuum cleaners (this is beneficial if we only have e.g., 3 of them— we can apply the algorithms in Section 4 to generate weighted trajectory set representations and compare them directly without going through the algorithm in Section 3.).

An interesting direction for future work is to relax the assumption that the MDP is known in advance. One direct way is to first learn the model (in model-based RL), then apply our algorithms in the learned MDP. The sub-optimality of the returned policy will then depend on both the estimation error of the model and the error introduced by our algorithms (which depends on the parameters in the learned model).

## Acknowledgements

This work was supported in part by the National Science Foundation under grants CCF-2212968 and ECCS-2216899 and by the Defense Advanced Research Projects Agency under cooperative agreement HR00112020003. The views expressed in this work do not necessarily reflect the position or the policy of the Government and no official endorsement should be inferred. Approved for public release; distribution is unlimited.

This project was supported in part by funding from the European Research Council (ERC) under the European Union's Horizon 2020 research and innovation program (grant agreement number 882396), by the Israel Science Foundation (grant number 993/17), Tel Aviv University Center for AI and Data Science (TAD), the Eric and Wendy Schmidt Fund, and the Yandex Initiative for Machine Learning at Tel Aviv University.

We would like to thank all anonymous reviewers, especially Reviewer Gnoo, for their constructive comments.

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

# A   Related Work

**Multi-objective sequential decision making** There is a long history of work on multi-objective sequential decision making (Roijers et al., 2013), with one key focus being the realization of efficient algorithms for approximating the Pareto front (Chatterjee et al., 2006; Chatterjee, 2007; Marinescu et al., 2017). Instead of finding a possibly optimal policy, we concentrate on specific user preferences and find a policy that is optimal for that specific user. Just like the ideal car for one person could be a Chevrolet Spark (small) and for another, it is a Ford Ranger (a truck).

In the context of multi-objective RL (Hayes et al., 2022), the goal can be formulated as one of learning a policy for which the average return vector belongs to a target set (hence the term "multi-criteria" RL), which existing work has treated as a stochastic game (Mannor and Shimkin, 2001, 2004). Other works seek to maximize (in expectation) a scalar version of the reward that may correspond to a weighted sum of the multiple objectives (Barrett and Narayanan, 2008; Chen et al., 2019) as we consider here, or a nonlinear function of the objectives (Cheung, 2019). Multi-objective online learning has also been studied, see (Mannor et al., 2014; Blum and Shao, 2020) for example.

The parameters that define this scalarization function (e.g., the relative objective weights) are often unknown and vary with the task setting or user. In this case, preference learning (Wirth et al., 2017a) is commonly used to elicit the value of these parameters. Doumpos and Zopounidis (2007) describe an approach to eliciting a user's relative weighting in the context of multi-objective decision-making. Bhatia et al. (2020) learn preferences over multiple objectives from pairwise queries using a game-theoretic approach to identify optimal randomized policies. In the context of RL involving both scalar and vector-valued objectives, user preference queries provide an alternative to learning from demonstrations, which may be difficult for people to provide (e.g., in the case of robots with high degrees-of-freedom), or explicit reward specifications (Cheng et al., 2011; Rothkopf and Dimitrakakis, 2011; Akrour et al., 2012; Wilson et al., 2012; Fürnkranz et al., 2012; Jain et al., 2015; Wirth et al., 2016; Christiano et al., 2017; Wirth et al., 2017b; Ibarz et al., 2018; Lee et al., 2021). These works typically assume that noisy human preferences over a pair of trajectories are correlated with the difference in their utilities (i.e., the reward acts as a latent term predictive of preference). Many contemporary methods estimate the latent reward by minimizing the cross-entropy loss between the reward-based predictions and the human-provided preferences (i.e., finding the reward that maximizes the likelihood of the observed preferences) (Christiano et al., 2017; Ibarz et al., 2018; Lee et al., 2021; Knox et al., 2022; Pacchiano et al., 2022).

Wilson et al. (2012) describe a Bayesian approach to policy learning whereby they query a user for their preference between a pair of trajectories and use these preferences to maintain a posterior distribution over the latent policy parameters. The task of choosing the most informative queries is challenging due to the continuous space of trajectories, and is generally NP-hard (Ailon, 2012). Instead, they assume access to a distribution over trajectories that accounts for their feasibility and relevance to the target policy, and they describe two heuristic approaches to selecting trajectory queries based on this distribution. Finally, Sadigh et al. (2017) describe an approach to active preference-based learning in continuous state and action spaces. Integral to their work is the ability to synthesize dynamically feasible trajectory queries. Biyik and Sadigh (2018) extend this approach to the batch query setting.

**Comparative feedback in other problems** Comparative feedback has been studied in other problems in learning theory, e.g., combinatorial functions (Balcan et al., 2016). One closely related problem is active ranking/learning using pairwise comparisons (Jamieson and Nowak, 2011; Kane et al., 2017; Saha et al., 2021; Yona et al., 2022). These works usually consider a given finite sample of points. Kane et al. (2017) implies a lower bound of the number of comparisons is linear in the cardinality of the set even if the points satisfy the linear structural constraint as we assume in this work. In our work, the points are value vectors generated by running different policies under the same MDP and thus have a specific structure. Besides, we allow comparison of policies not in the policy set. Thus, we are able to obtain the query complexity sublinear in the number of policies. Another related problem using comparative/preference feedback is dueling bandits that aim to learn through pairwise feedback (Ailon et al., 2014; Zoghi et al., 2014) (see also Bengs et al. (2021); Sui et al. (2018) for surveys), or more generally any subsetwise feedback (Sui et al., 2017; Saha and Gopalan, 2018, 2019; Ren et al., 2018). However, unlike dueling bandits, we consider noiseless comparative feedback.

# B  Proof of Lemma 1

**Lemma 1.** *When $\epsilon \leq \frac{v^*}{2k}$, we have $\frac{v^*}{\langle w^*, V^{\pi_1} \rangle} \leq 2k$, and $|\widehat{\alpha}_i - \alpha_i| \leq \frac{4k^2 \epsilon}{v^*}$ for every $i \in [d]$.*

*Proof.* We first show that, according to our algorithm (lines 1-6 of Algorithm 1), the returned $\pi_1$ satisfies that

$$\langle w^*, V^{\pi_1} \rangle \geq \max_{i \in [k]} \left\langle w^*, V^{\pi^{\mathbf{e}_i}} \right\rangle - \epsilon.$$

This can be proved by induction over $k$. In the base case of $k = 2$, it's easy to see that the returned $\pi_1$ satisfies the above inequality. Suppose the above inequality holds for any $k \leq n - 1$ and we prove that it will also hold for $k = n$. After running the algorithm over $j = 2, \ldots, k - 1$ (line 2), the returned policy $\pi_{e^*}$ satisfies that

$$\left\langle w^*, V^{\pi^{e^*}} \right\rangle \geq \max_{i \in [k-1]} \left\langle w^*, V^{\pi^{\mathbf{e}_i}} \right\rangle - \epsilon.$$

Then there are two cases,

- If $\left\langle w^*, V^{\pi^{e^*}} \right\rangle < \left\langle w^*, V^{\pi^{\mathbf{e}_k}} \right\rangle - \epsilon$, we will return $\pi_1 = \pi^{\mathbf{e}_k}$ and also, $\left\langle w^*, V^{\pi^{\mathbf{e}_k}} \right\rangle \geq \left\langle w^*, V^{\pi^{\mathbf{e}_i}} \right\rangle - \epsilon$ for all $i \in [k]$.

- If $\left\langle w^*, V^{\pi^{e^*}} \right\rangle \geq \left\langle w^*, V^{\pi^{\mathbf{e}_k}} \right\rangle - \epsilon$, then we will return $\pi^{e^*}$ and it satisfies $\left\langle w^*, V^{\pi^{e^*}} \right\rangle \geq \max_{i \in [k]} \left\langle w^*, V^{\pi^{\mathbf{e}_i}} \right\rangle - \epsilon$.

As $\pi^{\mathbf{e}_i}$ is the optimal personalized policy when the user's preference vector is $\mathbf{e}_i$, we have that

$$v^* = \left\langle w^*, V^{\pi^*} \right\rangle = \sum_{i=1}^{k} w_i^* \left\langle V^{\pi^*}, \mathbf{e}_i \right\rangle \leq \sum_{i=1}^{k} w_i^* \left\langle V^{\pi^{\mathbf{e}_i}}, \mathbf{e}_i \right\rangle \leq \left\langle w^*, \sum_{i=1}^{k} V^{\pi^{\mathbf{e}_i}} \right\rangle,$$

where the last inequality holds because the entries of $V^{\pi^{\mathbf{e}_i}}$ and $w^*$ are non-negative. Therefore, there exists $i \in [k]$ such that $\left\langle w^*, V^{\pi^{\mathbf{e}_i}} \right\rangle \geq \frac{1}{k} \left\langle w^*, V^{\pi^*} \right\rangle = \frac{1}{k} v^*$.

Then we have

$$\langle w^*, V^{\pi_1} \rangle \geq \max_{i \in [k]} \left\langle w^*, V^{\pi^{\mathbf{e}_i}} \right\rangle - \epsilon \geq \frac{1}{k} v^* - \epsilon \geq \frac{1}{2k} v^*,$$

when $\epsilon \leq \frac{v^*}{2k}$. By rearranging terms, we have $\frac{v^*}{\langle w^*, V^{\pi_1} \rangle} \leq 2k$.

By setting $C_\alpha = 2k$, we have $|\widehat{\alpha}_i - \alpha_i| \langle w^*, V^{\pi_1} \rangle \leq C_\alpha \epsilon = 2k\epsilon$ and thus, $|\widehat{\alpha}_i - \alpha_i| \leq \frac{4k^2 \epsilon}{v^*}$. $\qquad \square$

# C  Pseudo Code of Computation of the Basis Ratios

The pseudo code of searching $\widehat{\alpha}_i$'s is described in Algorithm 3.

**Algorithm 3** Computation of Basis Ratios

---

1: **input:** $(V^{\pi_1}, \ldots, V^{\pi_d})$ and $C_\alpha$
2: **for** $i = 1, \ldots, d-1$ **do**
3:     let $l = 0$, $h = 2C_\alpha$ and $\widehat{\alpha}_i = C_\alpha$
4:     **while** True **do**
5:       **if** $\widehat{\alpha}_i > 1$ **then**
6:         compare $\pi_1$ and $\frac{1}{\widehat{\alpha}_i}\pi_{i+1} + (1 - \frac{1}{\widehat{\alpha}_i})\pi_0$; **if** $\pi_1 \succ \frac{1}{\widehat{\alpha}_i}\pi_{i+1} + (1 - \frac{1}{\widehat{\alpha}_i})\pi_0$ **then** $h \leftarrow \widehat{\alpha}_i$,
        $\widehat{\alpha}_i \leftarrow \frac{l+h}{2}$; **if** $\pi_1 \prec \frac{1}{\widehat{\alpha}_i}\pi_{i+1} + (1 - \frac{1}{\widehat{\alpha}_i})\pi_0$ **then** $l \leftarrow \widehat{\alpha}_i$, $\widehat{\alpha}_i \leftarrow \frac{l+h}{2}$
7:       **else**
8:         compare $\pi_{i+1}$ and $\widehat{\alpha}_i\pi_1 + (1 - \widehat{\alpha}_i)\pi_0$; **if** $\widehat{\alpha}_i\pi_1 + (1 - \widehat{\alpha}_i)\pi_0 \succ \pi_{i+1}$ **then** $h \leftarrow \widehat{\alpha}_i$,
        $\widehat{\alpha}_i \leftarrow \frac{l+h}{2}$; **if** $\widehat{\alpha}_i\pi_1 + (1 - \widehat{\alpha}_i)\pi_0 \prec \pi_{i+1}$ **then** $l \leftarrow \widehat{\alpha}_i$, $\widehat{\alpha}_i \leftarrow \frac{l+h}{2}$
9:       **end if**
10:       **if** "indistinguishable" is returned **then**
11:         break
12:       **end if**
13:     **end while**
14: **end for**
15: **output:** $(\widehat{\alpha}_1, \ldots, \widehat{\alpha}_{d-1})$

---

# D   Proof of Theorem 1

**Theorem 1.** *Consider the algorithm of computing $\widehat{A}$ and any solution $\widehat{w}$ to $\widehat{A}x = \mathbf{e}_1$ and outputting the policy $\pi^{\widehat{w}} = \arg\max_{\pi \in \Pi} \langle \widehat{w}, V^\pi \rangle$, which is the optimal personalized policy for preference vector $\widehat{w}$. Then the output policy $\pi^{\widehat{w}}$ satisfying that $v^* - \left\langle w^*, V^{\pi^{\widehat{w}}} \right\rangle \leq \mathcal{O}\left( \left(\sqrt{k} + 1\right)^{d + \frac{14}{3}} \epsilon^{\frac{1}{3}} \right)$ using $\mathcal{O}(k \log(k/\epsilon))$ comparison queries.*

*Proof.* Theorem 1 follows by setting $\epsilon_\alpha = \frac{4k^2\epsilon}{v^*}$ and $C_\alpha = 2k$ as shown in Lemma 1 and combining the results of Lemma 2 and 3 with the triangle inequality.

Specifically, for any policy $\pi$, we have

$$|\langle \widehat{w}, V^\pi \rangle - \langle w', V^\pi \rangle| \leq \left| \langle \widehat{w}, V^\pi \rangle - \left\langle \widehat{w}^{(\delta)}, V^\pi \right\rangle \right| + \left| \left\langle \widehat{w}^{(\delta)}, V^\pi \right\rangle - \langle w', V^\pi \rangle \right|$$

$$\leq \mathcal{O}((\sqrt{k} + 1)^{d - d_\delta} C_\alpha (\frac{C_\alpha C_V^4 d_\delta^{\frac{3}{2}} \|w'\|_2^2 \epsilon_\alpha}{\delta^2} + \sqrt{k}\delta \|w'\|_2)) \quad (7)$$

$$\leq \mathcal{O}((\sqrt{k} + 1)^{d - d_\delta + 3} \|w'\|_2 (\frac{C_V^4 k^4 \|w'\|_2 \epsilon}{v^* \delta^2} + \delta)) .$$

Since $\|w'\| = \frac{\|w^*\|}{\langle w^*, V^{\pi_1} \rangle}$ and $\langle w^*, V^{\pi_1} \rangle \geq \frac{v^*}{2k}$ from Lemma 1, we derive

$$v^* - \left\langle w^*, V^{\pi^{\widehat{w}}} \right\rangle = \langle w^*, V^{\pi_1} \rangle \left( \left\langle w', V^{\pi^*} \right\rangle - \left\langle w', V^{\pi^{\widehat{w}}} \right\rangle \right)$$

$$\leq \langle w^*, V^{\pi_1} \rangle \left( \left\langle \widehat{w}, V^{\pi^*} \right\rangle - \left\langle \widehat{w}, V^{\pi^{\widehat{w}}} \right\rangle + \mathcal{O}((\sqrt{k} + 1)^{d - d_\delta + 3} \|w'\|_2 (\frac{C_V^4 k^4 \|w'\|_2 \epsilon}{v^* \delta^2} + \delta)) \right)$$

$$\leq \mathcal{O}\left( \langle w^*, V^{\pi_1} \rangle (\sqrt{k} + 1)^{d - d_\delta + 3} \|w'\|_2 (\frac{C_V^4 k^4 \|w'\|_2 \epsilon}{v^* \delta^2} + \delta) \right)$$

$$= \mathcal{O}\left( (\sqrt{k} + 1)^{d - d_\delta + 3} \|w^*\|_2 (\frac{C_V^4 k^5 \|w^*\|_2 \epsilon}{v^{*2} \delta^2} + \delta) \right)$$

$$= \mathcal{O}\left( (\frac{C_V^2 \|w^*\|_2^2}{v^*})^{\frac{2}{3}} (\sqrt{k} + 1)^{d + \frac{16}{3}} \epsilon^{\frac{1}{3}} \right) .$$

The first inequality follows from

$$\left\langle w', V^{\pi^*} \right\rangle - \left\langle w', V^{\pi^{\widehat{w}}} \right\rangle = \left\langle w', V^{\pi^*} \right\rangle - \left\langle w', V^{\pi^{\widehat{w}}} \right\rangle$$

$$+ \left( \left\langle \widehat{w}, V^{\pi^*} \right\rangle - \left\langle \widehat{w}, V^{\pi^*} \right\rangle \right) + \left( \left\langle \widehat{w}, V^{\pi^{\widehat{w}}} \right\rangle - \left\langle \widehat{w}, V^{\pi^{\widehat{w}}} \right\rangle \right),$$

and applying (7) twice- once for $\pi^*$ and once for $\pi^{\widehat{w}}$. The last inquality follows by setting $\delta = \left( \frac{C_V^4 k^5 \|w^*\|_2 \epsilon}{v^{*2}} \right)^{\frac{1}{3}}$. $\qquad\square$

## E  Proof of Lemma 2

**Lemma 2.** *If $|\widehat{\alpha}_i - \alpha_i| \le \epsilon_\alpha$ and $\alpha_i \le C_\alpha$ for all $i \in [d-1]$, for every $\delta \ge 4 C_\alpha^{\frac{2}{3}} C_V d^{\frac{1}{3}} \epsilon_\alpha^{\frac{1}{3}}$, we have*

$$\left| \left\langle \widehat{w}^{(\delta)}, V^\pi \right\rangle - \left\langle w', V^\pi \right\rangle \right| \le \mathcal{O}\left( \frac{C_\alpha C_V^4 d_\delta^{\frac{3}{2}} \|w'\|_2^2 \epsilon_\alpha}{\delta^2} + \sqrt{k}\delta \|w'\|_2 \right) \text{ for all } \pi, \text{ where } w' = \frac{w^*}{\langle w^*, V^{\pi_1} \rangle}.$$

To prove Lemma 2, we will first define a matrix, $A^{(\text{full})}$.

Given the output $(V^{\pi_1}, \ldots, V^{\pi_d})$ of Algorithm 1, we have $\text{rank}(\text{span}(\{V^{\pi_1}, \ldots, V^{\pi_{d_\delta}}\})) = d_\delta$.

Let $b_1, \ldots, b_{d-d_\delta}$ be a set of orthonormal vectors that are orthogonal to $\text{span}(V^{\pi_1}, \ldots, V^{\pi_{d_\delta}})$ and together with $V^{\pi_1}, \ldots, V^{\pi_{d_\delta}}$ form a basis for $\text{span}(\{V^\pi | \pi \in \Pi\})$.

We define $A^{(\text{full})} \in \mathbb{R}^{d \times k}$ as the matrix of replacing the last $d - d_\delta$ rows of $A$ with $b_1, \ldots, b_{d-d_\delta}$, i.e.,

$$A^{(\text{full})} = \begin{pmatrix} V^{\pi_1 \top} \\ (\alpha_1 V^{\pi_1} - V^{\pi_2})^\top \\ \vdots \\ (\alpha_{d_\delta - 1} V^{\pi_1} - V^{\pi_{d_\delta}})^\top \\ b_1^\top \\ \vdots \\ b_{d-d_\delta}^\top \end{pmatrix}.$$

**Observation 1.** *We have that $\text{span}(A^{(full)}) = \text{span}(\{V^\pi | \pi \in \Pi\})$ and $\text{rank}(A^{(full)}) = d$.*

**Lemma 5.** *For all $w \in \mathbb{R}^k$ satisfying $A^{(full)} w = \mathbf{e}_1$, we have $|w \cdot V^\pi - w' \cdot V^\pi| \le \sqrt{k}\delta \|w'\|_2$ for all $\pi$.*

We then show that there exists a $w \in \mathbb{R}^k$ satisfying $A^{(\text{full})} w = \mathbf{e}_1$ such that $\left| \widehat{w}^{(\delta)} \cdot V^\pi - w \cdot V^\pi \right|$ is small for all $\pi \in \Pi$.

**Lemma 6.** *If $|\widehat{\alpha}_i - \alpha_i| \le \epsilon_\alpha$ and $\alpha_i \le C_\alpha$ for all $i \in [d-1]$, for every $\delta \ge 4 C_\alpha^{\frac{2}{3}} C_V d^{\frac{1}{3}} \epsilon_\alpha^{\frac{1}{3}}$ there exists a $w \in \mathbb{R}^k$ satisfying $A^{(full)} w = \mathbf{e}_1$ s.t. $\left| \widehat{w}^{(\delta)} \cdot V^\pi - w \cdot V^\pi \right| \le \mathcal{O}\left( \frac{C_\alpha C_V^4 d_\delta^{\frac{3}{2}} \|w'\|_2^2 \epsilon_\alpha}{\delta^2} \right)$ for all $\pi$.*

We now derive Lemma 2 using the above two lemmas.

*Proof of Lemma 2.* Let $w$ be defined in Lemma 6. Then for any policy $\pi$, we have

$$\left| \widehat{w}^{(\delta)} \cdot V^\pi - w' \cdot V^\pi \right| \le \left| \widehat{w}^{(\delta)} \cdot V^\pi - w \cdot V^\pi \right| + |w \cdot V^\pi - w' \cdot V^\pi|$$

$$\le \mathcal{O}\left( \frac{C_\alpha C_V^4 d_\delta^{\frac{3}{2}} \|w'\|_2^2 \epsilon_\alpha}{\delta^2} + \sqrt{k}\delta \|w'\|_2 \right),$$

by applying Lemma 5 and 6. $\qquad\square$

## F  Proofs of Lemma 5 and Lemma 6

**Lemma 5.** *For all $w \in \mathbb{R}^k$ satisfying $A^{(full)} w = \mathbf{e}_1$, we have $|w \cdot V^\pi - w' \cdot V^\pi| \le \sqrt{k}\delta \|w'\|_2$ for all $\pi$.*

*Proof of Lemma 5.* Since $\text{span}(A^{(\text{full})}) = \text{span}(\{V^\pi | \pi \in \Pi\})$, for every policy $\pi$, the value vector can be represented as a linear combination of row vectors of $A^{(\text{full})}$, i.e., there exists $a = (a_1, \ldots, a_d) \in \mathbb{R}^d$ s.t.

$$V^\pi = \sum_{i=1}^{d} a_i A_i^{(\text{full})} = A^{(\text{full})\top} a . \tag{8}$$

Now, for any unit vector $\xi \in \text{span}(b_1, \ldots, b_{d-d_\delta})$, we have $\langle V^\pi, \xi \rangle \leq \sqrt{k}\delta$.

The reason is that at each round $d_\delta + 1$, we pick an orthonormal basis $\rho_1, \ldots, \rho_{k-d_\delta}$ of $\text{span}(V^{\pi_1}, \ldots, V^{\pi_{d_\delta}})^\perp$ (line 8 in Algorithm 1) and pick $u_{d_\delta+1}$ to be the one in which there exists a policy with the largest component as described in line 9. Hence, $|\langle \rho_j, V^\pi \rangle| \leq \delta$ for all $j \in [k - d_\delta]$.

It follows from Cauchy-Schwarz inequality that $\langle \xi, V^\pi \rangle = \sum_{j=1}^{k-d_\delta} \langle \xi, \rho_j \rangle \langle \rho_j, V^\pi \rangle \leq \sqrt{k}\delta$.

Combining with the observation that $b_1, \ldots, b_{d-d_\delta}$ are pairwise orthogonal and that each of them is orthogonal to $\text{span}(V^{\pi_1}, \ldots, V^{\pi_{d_\delta}})$ we have

$$\sum_{i=d_\delta+1}^{d} a_i^2 = \left| \left\langle V^\pi, \sum_{i=d_\delta+1}^{d} a_i b_{i-d_\delta} \right\rangle \right| \leq \sqrt{\sum_{i=d_\delta+1}^{d} a_i^2} \sqrt{k}\delta ,$$

which implies that

$$\sqrt{\sum_{i=d_\delta+1}^{d} a_i^2} \leq \sqrt{k}\delta . \tag{9}$$

Since $w'$ satisfies $Aw' = \mathbf{e}_1$, we have

$$A^{(\text{full})} w' = (1, 0, \ldots, 0, \langle b_1, w' \rangle, \ldots, \langle b_{d-d_\delta}, w' \rangle) .$$

For any $w \in \mathbb{R}^k$ satisfying $A^{(\text{full})} w = \mathbf{e}_1$, consider $\widetilde{w} = w + \sum_{i=1}^{d-d_\delta} \langle b_i, w' \rangle b_i$. Then we have $A^{(\text{full})} \widetilde{w} = A^{(\text{full})} w'$.

Thus, applying (8) twice, we get

$$\widetilde{w} \cdot V^\pi = \widetilde{w}^\top A^{(\text{full})\top} a = w'^\top A^{(\text{full})\top} a = w' \cdot V^\pi .$$

Hence,

$$|w \cdot V^\pi - w' \cdot V^\pi| = |w \cdot V^\pi - \widetilde{w} \cdot V^\pi| \stackrel{(a)}{=} \left| \sum_{i=1}^{d} a_i (w - \widetilde{w}) \cdot A_i^{(\text{full})} \right|$$

$$= \left| \sum_{i=d_\delta+1}^{d} a_i \langle b_{i-d_\delta}, w' \rangle \right| \stackrel{(b)}{\leq} \sqrt{\sum_{i=d_\delta+1}^{d} a_i^2} \|w'\|_2 \stackrel{(c)}{\leq} \sqrt{k}\delta \|w'\|_2 ,$$

where Eq (a) follows from (8), inequality (b) from Cauchy-Schwarz, and inequality (c) from applying (9).

$\square$

**Lemma 6.** *If $|\widehat{\alpha}_i - \alpha_i| \leq \epsilon_\alpha$ and $\alpha_i \leq C_\alpha$ for all $i \in [d-1]$, for every $\delta \geq 4C_\alpha^{\frac{2}{3}} C_V d^{\frac{1}{3}} \epsilon_\alpha^{\frac{1}{3}}$ there exists a $w \in \mathbb{R}^k$ satisfying $A^{(\text{full})} w = \mathbf{e}_1$ s.t. $\left| \widehat{w}^{(\delta)} \cdot V^\pi - w \cdot V^\pi \right| \leq \mathcal{O}(\frac{C_\alpha C_V^4 d^{\frac{3}{2}} \|w'\|_2^2 \epsilon_\alpha}{\delta^2})$ for all $\pi$.*

Before proving Lemma 6, we introduce some notations and a claim.

- For any $x, y \in \mathbb{R}^k$, let $\theta(x, y)$ denotes the angle between $x$ and $y$.
- For any subspace $U \subset \mathbb{R}^k$, let $\theta(x, U) := \min_{y \in U} \theta(x, y)$.
- For any two subspaces $U, U' \subset \mathbb{R}^k$, we define $\theta(U, U')$ as $\theta(U, U') = \max_{x \in U} \min_{y \in U'} \theta(x, y)$.

- For any matrix $M$, let $M_i$ denote the $i$-th row vector of $M$, $M_{i:j}$ denote the submatrix of $M$ composed of rows $i, i+1, \ldots, j$, and $M_{i:}$ denote the submatrix composed of all rows $j \geq i$.
- Let $\mathrm{span}(M)$ denote the span of the rows of $M$.

Recall that $\widehat{A} \in \mathbb{R}^{d \times k}$ is defined as

$$
\widehat{A} = \begin{pmatrix} V^{\pi_1 \top} \\ (\widehat{\alpha}_1 V^{\pi_1} - V^{\pi_2})^\top \\ \vdots \\ (\widehat{\alpha}_{d-1} V^{\pi_1} - V^{\pi_d})^\top \end{pmatrix},
$$

which is the approximation of matrix $A \in \mathbb{R}^{d \times k}$ defined by true values of $\alpha_i$, i.e.,

$$
A = \begin{pmatrix} V^{\pi_1 \top} \\ (\alpha_1 V^{\pi_1} - V^{\pi_2})^\top \\ \vdots \\ (\alpha_{d-1} V^{\pi_1} - V^{\pi_d})^\top \end{pmatrix}.
$$

We denote by $\widehat{A}^{(\delta)} = \widehat{A}_{1:d_\delta}, A^{(\delta)} = A_{1:d_\delta} \in \mathbb{R}^{d_\delta \times k}$ the sub-matrices comprised of the first $d_\delta$ rows of $\widehat{A}$ and $A$ respectively.

**Claim 1.** *If $|\widehat{\alpha}_i - \alpha_i| \leq \epsilon_\alpha$ and $\alpha_i \leq C_\alpha$ for all $i \in [d-1]$, for every $\delta \geq 4 C_\alpha^{\frac{2}{3}} C_V d^{\frac{1}{3}} \epsilon_\alpha^{\frac{1}{3}}$, we have*

$$
\theta(\mathrm{span}(A_{2:}^{(\delta)}), \mathrm{span}(\widehat{A}_{2:}^{(\delta)})) \leq \eta_{\epsilon_\alpha, \delta}, \tag{10}
$$

*and*

$$
\theta(\mathrm{span}(\widehat{A}_{2:}^{(\delta)}), \mathrm{span}(A_{2:}^{(\delta)})) \leq \eta_{\epsilon_\alpha, \delta}, \tag{11}
$$

*where $\eta_{\epsilon_\alpha, \delta} = \frac{4 C_\alpha C_V^2 d_\delta \epsilon_\alpha}{\delta^2}$.*

To prove the above claim, we use the following lemma by Balcan et al. (2015).

**Lemma 7** (Lemma 3 of Balcan et al. (2015)). *Let $U_l = \mathrm{span}(\xi_1, \ldots, \xi_l)$ and $\widehat{U}_l = \mathrm{span}(\widehat{\xi}_1, \ldots, \widehat{\xi}_l)$. Let $\epsilon_{acc}, \gamma_{new} \geq 0$ and $\epsilon_{acc} \leq \gamma_{new}^2/(10l)$, and assume that $\theta(\widehat{\xi}_i, \widehat{U}_{i-1}) \geq \gamma_{new}$ for $i = 2, \ldots, l$, and that $\theta(\xi_i, \widehat{\xi}_i) \leq \epsilon_{acc}$ for $i = 1, \ldots, l$.*

*Then,*

$$
\theta(U_l, \widehat{U}_l) \leq 2l \frac{\epsilon_{acc}}{\gamma_{new}}.
$$

*Proof of Claim 1.* For all $2 \leq i \leq d_\delta$, we have that

$$
\theta(\widehat{A}_i^{(\delta)}, \mathrm{span}(\widehat{A}_{2:i-1}^{(\delta)})) \geq \theta(\widehat{A}_i^{(\delta)}, \mathrm{span}(\widehat{A}_{1:i-1}^{(\delta)})) \geq \sin(\theta(\widehat{A}_i^{(\delta)}, \mathrm{span}(\widehat{A}_{1:i-1}^{(\delta)})))
$$

$$
\overset{(a)}{\geq} \frac{\left| \widehat{A}_i^{(\delta)} \cdot u_i \right|}{\left\| \widehat{A}_i^{(\delta)} \right\|_2} \overset{(b)}{\geq} \frac{\delta}{\left\| \widehat{A}_i^{(\delta)} \right\|_2} = \frac{\delta}{\left\| \widehat{\alpha}_{i-1} V^{\pi_1} - V^{\pi_i} \right\|_2} \geq \frac{\delta}{(C_\alpha + 1) C_V},
$$

where Ineq (a) holds as $u_i$ is orthogonal to $\mathrm{span}(\widehat{A}_{1:i-1}^{(\delta)})$ according to line 8 of Algorithm 1 and Ineq (b) holds due to $\left| \widehat{A}_i^{(\delta)} \cdot u_i \right| = |V^{\pi_i} \cdot u_i| \geq \delta$. The last inequality holds due to $\left\| \widehat{\alpha}_{i-1} V^{\pi_1} - V^{\pi_i} \right\|_2 \leq \widehat{\alpha}_{i-1} \left\| V^{\pi_1} \right\|_2 + \left\| V_i \right\|_2 \leq (C_\alpha + 1) C_V$.

Similarly, we also have

$$
\theta(A_i^{(\delta)}, \mathrm{span}(A_{2:i-1}^{(\delta)})) \geq \frac{\delta}{(C_\alpha + 1) C_V}.
$$

We continue by decomposing $V^{\pi_i}$ in the direction of $V^{\pi_1}$ and the direction perpendicular to $V^{\pi_1}$.

For convince, we denote $v_i^\parallel := V^{\pi_i} \cdot \frac{V^{\pi_1}}{\|V^{\pi_1}\|_2}$, $V_i^\parallel := v_i^\parallel \frac{V^{\pi_1}}{\|V^{\pi_1}\|_2}$, $V_i^\perp := V^{\pi_i} - V_i^\parallel$ and $v_i^\perp := \left\| V_i^\perp \right\|_2$.

Then we have

$$\theta(A_i^{(\delta)}, \widehat{A}_i^{(\delta)}) = \theta(\alpha_{i-1}V^{\pi_1} - V^{\pi_i}, \widehat{\alpha}_{i-1}V^{\pi_1} - V^{\pi_i}) = \theta(\alpha_{i-1}V^{\pi_1} - V_i^{\|} - V_i^{\perp}, \widehat{\alpha}_{i-1}V^{\pi_1} - V_i^{\|} - V_i^{\perp}).$$

If $(\widehat{\alpha}_{i-1}V^{\pi_1} - V_i^{\|}) \cdot (\alpha_{i-1}V^{\pi_1} - V_i^{\|}) \geq 0$, i.e., $\widehat{\alpha}_{i-1}V^{\pi_1} - V_i^{\|}$ and $\alpha_{i-1}V^{\pi_1} - V_i^{\|}$ are in the same direction, then

$$\theta(A_i^{(\delta)}, \widehat{A}_i^{(\delta)}) = \left| \arctan \frac{\left\| \widehat{\alpha}_{i-1}V^{\pi_1} - V_i^{\|} \right\|_2}{v_i^{\perp}} - \arctan \frac{\left\| \alpha_{i-1}V^{\pi_1} - V_i^{\|} \right\|_2}{v_i^{\perp}} \right|$$

$$\leq \left| \frac{\left\| \widehat{\alpha}_{i-1}V^{\pi_1} - V_i^{\|} \right\|_2}{v_i^{\perp}} - \frac{\left\| \alpha_{i-1}V^{\pi_1} - V_i^{\|} \right\|_2}{v_i^{\perp}} \right| \tag{12}$$

$$= \frac{|\widehat{\alpha}_{i-1} - \alpha_{i-1}| \, \|V^{\pi_1}\|_2}{v_i^{\perp}}$$

$$\leq \frac{\epsilon_\alpha C_V}{\delta}, \tag{13}$$

where Ineq (12) follows from the fact that the derivative of $\arctan$ is at most 1, i.e., $\frac{\partial \arctan x}{\partial x} = \lim_{a \to x} \frac{\arctan a - \arctan x}{a - x} = \frac{1}{1+x^2} \leq 1$.

Inequality (13) holds since $v_i^{\perp} \geq |\langle V^{\pi_i}, u_i \rangle| \geq \delta$.

If $(\widehat{\alpha}_{i-1}V^{\pi_1} - V_i^{\|}) \cdot (\alpha_{i-1}V^{\pi_1} - V_i^{\|}) < 0$, i.e., $\widehat{\alpha}_{i-1}V^{\pi_1} - V_i^{\|}$ and $\alpha_{i-1}V^{\pi_1} - V_i^{\|}$ are in the opposite directions, then we have $\left\| \widehat{\alpha}_{i-1}V^{\pi_1} - V_i^{\|} \right\|_2 + \left\| \alpha_{i-1}V^{\pi_1} - V_i^{\|} \right\|_2 = \|(\widehat{\alpha}_{i-1} - \alpha_{i-1})V^{\pi_1}\|_2 \leq \epsilon_\alpha \|V^{\pi_1}\|_2$.

Similarly, we have

$$\theta(\widehat{A}_i^{(\delta)}, A_i^{(\delta)}) = \left| \arctan \frac{\left\| \widehat{\alpha}_{i-1}V^{\pi_1} - V_i^{\|} \right\|_2}{v_i^{\perp}} + \arctan \frac{\left\| \alpha_{i-1}V^{\pi_1} - V_i^{\|} \right\|_2}{v_i^{\perp}} \right|$$

$$\leq \left| \frac{\left\| \widehat{\alpha}_{i-1}V^{\pi_1} - V_i^{\|} \right\|_2}{v_i^{\perp}} + \frac{\left\| \alpha_{i-1}V^{\pi_1} - V_i^{\|} \right\|_2}{v_i^{\perp}} \right|$$

$$\leq \frac{\epsilon_\alpha \|V^{\pi_1}\|_2}{|v_i^{\perp}|}$$

$$\leq \frac{\epsilon_\alpha C_V}{\delta}.$$

By applying Lemma 7 with $\epsilon_{\text{acc}} = \frac{\epsilon_\alpha C_V}{\delta}$, $\gamma_{\text{new}} = \frac{\delta}{(C_\alpha + 1)C_V}$, $(\xi_i, \widehat{\xi}_i) = (A_{i+1}, \widehat{A}_{i+1})$ (and $(\xi_i, \widehat{\xi}_i) = (\widehat{A}_{i+1}, A_{i+1})$), we have that when $\delta \geq 10^{\frac{1}{3}}(C_\alpha + 1)^{\frac{2}{3}}C_V d_\delta^{\frac{1}{3}}\epsilon_\alpha^{\frac{1}{3}}$,

$$\theta(\text{span}(A_{2:}^{(\delta)}), \text{span}(\widehat{A}_{2:}^{(\delta)})) \leq \frac{2d_\delta(C_\alpha + 1)C_V^2 \epsilon_\alpha}{\delta^2} = \eta_{\epsilon_\alpha, \delta},$$

and

$$\theta(\text{span}(\widehat{A}_{2:}^{(\delta)}), \text{span}(A_{2:}^{(\delta)})) \leq \eta_{\epsilon_\alpha, \delta}.$$

This completes the proof of Claim 1 since $C_\alpha \geq 1$. $\qquad\square$

*Proof of Lemma 6.* Recall that $\widehat{w}^{(\delta)} = \arg\min_{\widehat{A}^{(\delta)}x = \mathbf{e}_1} \|x\|_2$ is the minimum norm solution to $\widehat{A}^{(\delta)}x = \mathbf{e}_1$.

Thus, $\langle \widehat{w}^{(\delta)}, b_i \rangle = 0$ for all $i \in [d - d_\delta]$.

Let $\lambda_1, \ldots, \lambda_{d_\delta - 1}$ be any orthonormal basis of $\text{span}(A_{2:}^{(\delta)})$.

We construct a vector $w$ satisfying $A^{\text{(full)}}w = \mathbf{e}_1$ by removing $\widehat{w}^{(\delta)}$'s component in $\text{span}(A_{2:}^{(\delta)})$ and rescaling.

Formally,

$$w := \frac{\widehat{w}^{(\delta)} - \sum_{i=1}^{d_\delta - 1} \left\langle \widehat{w}^{(\delta)}, \lambda_i \right\rangle \lambda_i}{1 - V^{\pi_1} \cdot \left( \sum_{i=1}^{d_\delta - 1} \left\langle \widehat{w}^{(\delta)}, \lambda_i \right\rangle \lambda_i \right)} . \tag{14}$$

It is direct to verify that $A_1 \cdot w = V^{\pi_1} \cdot w = 1$ and $A_i \cdot w = 0$ for $i = 2, \ldots, d_\delta$. As a result, $A^{(\delta)}w = \mathbf{e}_1$.

Combining with the fact that $\widehat{w}^{(\delta)}$ has zero component in $b_i$ for all $i \in [d - d_\delta]$, we have $A^{\text{(full)}}w = \mathbf{e}_1$.

According to Claim 1, we have

$$\theta(\text{span}(A_{2:}^{(\delta)}), \text{span}(\widehat{A}_{2:}^{(\delta)})) \le \eta_{\epsilon_\alpha, \delta}.$$

Thus, there exist unit vectors $\widetilde{\lambda}_1, \ldots, \widetilde{\lambda}_{d_\delta - 1} \in \text{span}(\widehat{A}_{2:}^{(\delta)})$ such that $\theta(\lambda_i, \widetilde{\lambda}_i) \le \eta_{\epsilon_\alpha, \delta}$.

Since $\widehat{A}^{(\delta)}\widehat{w}^{(\delta)} = \mathbf{e}_1$, we have $\widehat{w}^{(\delta)} \cdot \widetilde{\lambda}_i = 0$ for all $i = 1, \ldots, d_\delta - 1$, and therefore,

$$\left| \widehat{w}^{(\delta)} \cdot \lambda_i \right| = \left| \widehat{w}^{(\delta)} \cdot (\lambda_i - \widetilde{\lambda}_i) \right| \le \left\| \widehat{w}^{(\delta)} \right\|_2 \eta_{\epsilon_\alpha, \delta} .$$

This implies that for any policy $\pi$,

$$\left| V^\pi \cdot \sum_{i=1}^{d_\delta - 1} (\widehat{w}^{(\delta)} \cdot \lambda_i)\lambda_i \right| \le \|V^\pi\|_2 \sqrt{d_\delta} \left\| \widehat{w}^{(\delta)} \right\|_2 \eta_{\epsilon_\alpha, \delta} \le C_V \sqrt{d_\delta} \left\| \widehat{w}^{(\delta)} \right\|_2 \eta_{\epsilon_\alpha, \delta} .$$

Denote by $\gamma = V^{\pi_1} \cdot \left( \sum_{i=1}^{d_\delta - 1} \left\langle \widehat{w}^{(\delta)}, \lambda_i \right\rangle \lambda_i \right)$, which is no greater than $C_V \sqrt{d_\delta} \left\| \widehat{w}^{(\delta)} \right\|_2 \eta_{\epsilon_\alpha, \delta}$.

We have that

$$\begin{aligned}
\left| \widehat{w}^{(\delta)} \cdot V^\pi - w \cdot V^\pi \right| &\le \left| \widehat{w}^{(\delta)} \cdot V^\pi - \frac{1}{1-\gamma} \widehat{w}^{(\delta)} \cdot V^\pi \right| + \left| \frac{1}{1-\gamma} \widehat{w}^{(\delta)} \cdot V^\pi - w \cdot V^\pi \right| \\
&\le \frac{\gamma \left\| \widehat{w}^{(\delta)} \right\|_2 C_V}{1-\gamma} + \frac{1}{1-\gamma} \left| \sum_{i=1}^{d_\delta - 1} (\widehat{w}^{(\delta)} \cdot \lambda_i)\lambda_i \cdot V^\pi \right| \\
&\le \frac{\gamma \left\| \widehat{w}^{(\delta)} \right\|_2 C_V}{1-\gamma} + \frac{C_V \sqrt{d_\delta} \left\| \widehat{w}^{(\delta)} \right\|_2 \eta_{\epsilon_\alpha, \delta}}{1-\gamma} \\
&= 2(C_V \left\| \widehat{w}^{(\delta)} \right\|_2 + 1)C_V \sqrt{d_\delta} \left\| \widehat{w}^{(\delta)} \right\|_2 \eta_{\epsilon_\alpha, \delta}, \tag{15}
\end{aligned}$$

where the last equality holds when $C_V \sqrt{d_\delta} \left\| \widehat{w}^{(\delta)} \right\|_2 \eta_{\epsilon_\alpha, \delta} \le \frac{1}{2}$.

Now we show that $\left\| \widehat{w}^{(\delta)} \right\|_2 \le C \left\| w' \right\|_2$ for some constant $C$.

Since $\widehat{w}^{(\delta)}$ is the minimum norm solution to $\widehat{A}^{(\delta)}x = \mathbf{e}_1$, we will construct another solution to $\widehat{A}^{(\delta)}x = \mathbf{e}_1$, denoted by $\widehat{w}_0$ in a similliar manner to the construction in Eq (14), and show that $\|\widehat{w}_0\|_2 \le C \|w'\|$.

Let $\xi_1, \ldots, \xi_{d_\delta - 1}$ be any orthonormal basis of $\text{span}(\widehat{A}_{2:}^{(\delta)})$.

We construct a $\widehat{w}_0$ s.t. $\widehat{A}^{(\delta)}\widehat{w}_0 = \mathbf{e}_1$ by removing the component of $w'$ in $\text{span}(\widehat{A}_{2:}^{(\delta)})$ and rescaling. Specifically, let

$$\widehat{w}_0 = \frac{w' - \sum_{i=1}^{d_\delta - 1} \left\langle w', \xi_i \right\rangle \xi_i}{1 - \left\langle V^{\pi_1}, \left( \sum_{i=1}^{d_\delta - 1} \left\langle w', \xi_i \right\rangle \xi_i \right) \right\rangle} . \tag{16}$$

Since $A^{(\delta)}w' = \mathbf{e}_1$, it directly follows that $\left\langle \widehat{A}_1, \widehat{w}_0 \right\rangle = \langle V^{\pi_1}, \widehat{w}_0 \rangle = 1$ and that $\left\langle \widehat{A}_i, \widehat{w}_0 \right\rangle = 0$ for $i = 2, \ldots, d_\delta$, i.e., $\widehat{A}^{(\delta)}\widehat{w}_0 = \mathbf{e}_1$.

Since Claim 1 implies that $\theta(\text{span}(\widehat{A}_{2:}^{(\delta)}), \text{span}(A_{2:}^{(\delta)})) \leq \eta_{\epsilon_\alpha, \delta}$, there exist unit vectors $\widetilde{\xi}_1, \ldots, \widetilde{\xi}_{d_\delta - 1} \in \text{span}(A_{2:}^{(\delta)})$ such that $\theta(\xi_i, \widetilde{\xi}_i) \leq \eta_{\epsilon_\alpha, \delta}$.

As $w'$ has zero component in $\text{span}(A_{2:}^{(\delta)})$, $w'$ should have a small component in $\text{span}(\widehat{A}_{2:}^{(\delta)})$.

In particular,

$$\left| \langle w', \xi_i \rangle \right| = \left| \left\langle w', \xi_i - \widetilde{\xi}_i \right\rangle \right| \leq \|w'\|_2 \, \eta_{\epsilon_\alpha, \delta} \,,$$

which implies that

$$\left\| \sum_{i=1}^{d_\delta - 1} \langle w', \xi_i \rangle \, \xi_i \right\|_2 \leq \sqrt{d_\delta} \, \|w'\|_2 \, \eta_{\epsilon_\alpha, \delta} \,.$$

Hence

$$\left| \left\langle V^{\pi_1}, \left( \sum_{i=1}^{d_\delta - 1} \langle w', \xi_i \rangle \, \xi_i \right) \right\rangle \right| \leq C_V \sqrt{d_\delta} \, \|w'\|_2 \, \eta_{\epsilon_\alpha, \delta} \,.$$

As a result, $\|\widehat{w}_0\|_2 \leq \frac{3}{2} \|w'\|_2$ when $C_V \sqrt{d_\delta} \|w'\|_2 \, \eta_{\epsilon_\alpha, \delta} \leq \frac{1}{3}$, which is true when $\epsilon_\alpha$ is small enough.

According to Lemma 1, $\epsilon_\alpha \leq \frac{4k^2 \epsilon}{v^*}$, $\epsilon_\alpha \to 0$ as $\epsilon \to 0$.

Thus, we have $\left\| \widehat{w}^{(\delta)} \right\|_2 \leq \|\widehat{w}_0\|_2 \leq \frac{3}{2} \|w'\|_2$ and $C_V \sqrt{d_\delta} \left\| \widehat{w}^{(\delta)} \right\|_2 \eta_{\epsilon_\alpha, \delta} \leq \frac{1}{2}$.

Combined with Eq (15), we get

$$\left| \widehat{w}^{(\delta)} \cdot V^\pi - w \cdot V^\pi \right| = \mathcal{O} \left( (C_V \|w'\|_2 + 1) C_V \sqrt{d_\delta} \|w'\|_2 \, \eta_{\epsilon_\alpha, \delta} \right) \,.$$

Since $C_V \|w'\|_2 \geq |\langle V^{\pi_1}, w' \rangle| = 1$, by taking $\eta_{\epsilon_\alpha, \delta} = \frac{4 C_\alpha C_V^2 d_\delta \epsilon_\alpha}{\delta^2}$ into the above equation, we have

$$\left| \widehat{w}^{(\delta)} \cdot V^\pi - w \cdot V^\pi \right| = \mathcal{O} \left( \frac{C_\alpha C_V^4 d_\delta^{\frac{3}{2}} \|w'\|_2^2 \epsilon_\alpha}{\delta^2} \right) \,,$$

which completes the proof. $\qquad \square$

## G   Proof of Lemma 3

**Lemma 3.** *If* $|\widehat{\alpha}_i - \alpha_i| \leq \epsilon_\alpha$ *and* $\alpha_i \leq C_\alpha$ *for all* $i \in [d-1]$, *for every policy* $\pi$ *and every* $\delta \geq 4 C_\alpha^{\frac{2}{3}} C_V d^{\frac{1}{3}} \epsilon_\alpha^{\frac{1}{3}}$, *we have* $\left| \widehat{w} \cdot V^\pi - \widehat{w}^{(\delta)} \cdot V^\pi \right| \leq \mathcal{O}((\sqrt{k} + 1)^{d - d_\delta} C_\alpha \epsilon^{(\delta)})$, *where* $\epsilon^{(\delta)} = \frac{C_\alpha C_V^4 d_\delta^{\frac{3}{2}} \|w'\|_2^2 \epsilon_\alpha}{\delta^2} + \sqrt{k} \delta \|w'\|_2$ *is the upper bound in Lemma 2.*

*Proof.* Given the output $(V^{\pi_1}, \ldots, V^{\pi_d})$ of Algorithm 1, we have $\text{rank}(\text{span}(\{V^{\pi_1}, \ldots, V^{\pi_{d_\delta}}\})) = d_\delta$.

For $i = d_\delta + 1, \ldots, d$, let $\psi_i$ be the normalized vector of $V^{\pi_i}$'s projection into $\text{span}(V^{\pi_1}, \ldots, V^{\pi_{i-1}})^\perp$ with $\|\psi_i\|_2 = 1$.

Then we have that $\text{span}(V^{\pi_1}, \ldots, V^{\pi_{i-1}}, \psi_i) = \text{span}(V^{\pi_1}, \ldots, V^{\pi_i})$ and that $\{\psi_i | i = d_\delta + 1, \ldots, d\}$ are orthonormal.

For every policy $\pi$, the value vector can be represented as a linear combination of $\widehat{A}_1, \ldots, \widehat{A}_{d_\delta}, \psi_{d_\delta + 1}, \ldots, \psi_d$, i.e., there exists a unique $a = (a_1, \ldots, a_d) \in \mathbb{R}^d$ s.t. $V^\pi = \sum_{i=1}^{d_\delta} a_i \widehat{A}_i + \sum_{i=d_\delta + 1}^{d} a_i \psi_i$.

Since $\psi_i$ is orthogonal to $\psi_j$ for all $j \neq i$ and $\psi_i$ is are orthogonal to $\text{span}(\widehat{A}_1, \ldots, \widehat{A}_{d_\delta})$, we have $a_i = \langle V^\pi, \psi_i \rangle$ for $i \geq d_\delta + 1$.

This implies that

$$\left| \langle \widehat{w}, V^\pi \rangle - \left\langle \widehat{w}^{(\delta)}, V^\pi \right\rangle \right| \le \underbrace{\left| \sum_{i=1}^{d_\delta} a_i \left( \left\langle \widehat{w}, \widehat{A}_i \right\rangle - \left\langle w^{(\delta)}, \widehat{A}_i \right\rangle \right) \right|}_{(a)} + \underbrace{\left| \sum_{i=d_\delta+1}^{d} a_i \langle \widehat{w}, \psi_i \rangle \right|}_{(b)}$$

$$+ \underbrace{\left| \sum_{i=d_\delta+1}^{d} a_i \left\langle \widehat{w}^{(\delta)}, \psi_i \right\rangle \right|}_{(c)} .$$

Since $\widehat{A}^{(\delta)} \widehat{w} = \widehat{A}^{(\delta)} \widehat{w}^{(\delta)} = \mathbf{e}_1$, we have term $(a) = 0$.

We move on to bound term (c).

Note that the vectors $\{\psi_i | i = d_\delta+1, \ldots, d\}$ are orthogonal to $\mathrm{span}(V^{\pi_1}, \ldots, V^{\pi_{d_\delta}})$ and that together with $V^{\pi_1}, \ldots, V^{\pi_{d_\delta}}$ they form a basis for $\mathrm{span}(\{V^\pi | \pi \in \Pi\})$.

Thus, we can let $b_i$ in the proof of Lemma 2 be $\psi_{i+d_\delta}$.

In addition, all the properties of $\{b_i | i \in [d - d_\delta]\}$ also apply to $\{\psi_i | i = d_\delta + 1, \ldots, d\}$ as well.

Hence, similarly to Eq (9),

$$\sqrt{\sum_{i=d_\delta+1}^{d} a_i^2} \le \sqrt{k}\delta .$$

Consequentially, we can bound term $(c)$ is by

$$(c) \le \sqrt{k}\delta \left\| \widehat{w}^{(\delta)} \right\|_2 = \frac{3}{2} \sqrt{k}\delta \left\| w' \right\|_2$$

since $\left\| \widehat{w}^{(\delta)} \right\|_2 \le \frac{3}{2} \left\| w' \right\|_2$ when $C_V \sqrt{d_\delta} \left\| w' \right\|_2 \eta_{\epsilon_\alpha, \delta} \le \frac{1}{3}$ as discussed in the proof of Lemma 6.

Now all is left is to bound term (b).

We cannot bound term (b) in the same way as that of term (c) because $\left\| \widehat{w} \right\|_2$ is not guaranteed to be bounded by $\left\| w' \right\|_2$.

For $i = d_\delta + 1, \ldots, d$, we define

$$\epsilon_i := \left| \left\langle \psi_i, \widehat{A}_i \right\rangle \right| .$$

For any $i, j = d_\delta + 1, \ldots, d$, $\psi_i$ is perpendicular to $V^{\pi_1}$, thus $\left| \left\langle \psi_i, \widehat{A}_j \right\rangle \right| = |\langle \psi_i, \widehat{\alpha}_{j-1} V^{\pi_1} - V^{\pi_j} \rangle| = |\langle \psi_i, V^{\pi_j} \rangle|$. Especially, we have

$$\epsilon_i = \left| \left\langle \psi_i, \widehat{A}_i \right\rangle \right| = |\langle \psi_i, V^{\pi_i} \rangle| .$$

Let $\widehat{A}_i^{\|} := \widehat{A}_i - \sum_{j=d_\delta+1}^{d} \left\langle \widehat{A}_i, \psi_j \right\rangle \psi_j$ denote $\widehat{A}_i$'s projection into $\mathrm{span}(\widehat{A}_1, \ldots, \widehat{A}_{d_\delta})$.

Since $\widehat{A}_i$ has zero component in direction $\psi_j$ for $j > i$, we have $\widehat{A}_i^{\|} = \widehat{A}_i - \sum_{j=d_\delta+1}^{i} \left\langle \widehat{A}_i, \psi_j \right\rangle \psi_j$.

Then, we have

$$0 = \left\langle \widehat{w}, \widehat{A}_i \right\rangle = \widehat{w} \cdot \widehat{A}_i^{\|} + \widehat{w} \cdot \sum_{j=d_\delta+1}^{i} \left\langle \widehat{A}_i, \psi_j \right\rangle \psi_j = \widehat{w} \cdot \widehat{A}_i^{\|} - \sum_{j=d_\delta+1}^{i} \langle V^{\pi_i}, \psi_j \rangle \langle \widehat{w}, \psi_j \rangle ,$$

where the first equation holds due to $\widehat{A}\widehat{w} = \mathbf{e}_1$.

By rearranging terms, we have

$$\langle V^{\pi_i}, \psi_i \rangle \langle \widehat{w}, \psi_i \rangle = \widehat{w} \cdot \widehat{A}_i^{\|} - \sum_{j=d_\delta+1}^{i-1} \langle V^{\pi_i}, \psi_j \rangle \langle \widehat{w}, \psi_j \rangle . \tag{17}$$

Recall that at iteration $j$ of Algorithm 1, in line 8 we pick an orthonormal basis $\rho_1, \ldots, \rho_{k+1-j}$ of $\mathrm{span}(V^{\pi_1}, \ldots, V^{\pi_{j-1}})^\perp$. Since $\psi_j$ is in $\mathrm{span}(V^{\pi_1}, \ldots, V^{\pi_{j-1}})^\perp$ according to the definition of $\psi_j$, $|\langle V^{\pi_i}, \psi_j \rangle|$ is no greater then the norm of $V^{\pi_i}$'s projection into $\mathrm{span}(V^{\pi_1}, \ldots, V^{\pi_{j-1}})^\perp$.

Therefore, we have

$$|\langle V^{\pi_i}, \psi_j \rangle| \leq \sqrt{k} \max_{l \in [k+1-j]} |\langle V^{\pi_i}, \rho_l \rangle| \overset{(d)}{\leq} \sqrt{k} \max_{l \in [k+1-j]} \max(\left| \left\langle V^{\pi^{\rho_l}}, \rho_l \right\rangle \right|, \left| \left\langle V^{\pi^{-\rho_l}}, -\rho_l \right\rangle \right|)$$

$$\overset{(e)}{=} \sqrt{k} |\langle V^{\pi_j}, u_j \rangle| \overset{(f)}{\leq} \sqrt{k} |\langle V^{\pi_j}, \psi_j \rangle| = \sqrt{k} \epsilon_j , \tag{18}$$

where inequality (d) holds because $\pi^{\rho_l}$ is the optimal personalized policy with respect to the preference vector $\rho_l$, and Equation (e) holds due to the definition of $u_j$ (line 9 of Algorithm 1). Inequality (f) holds since $\langle V^{\pi_j}, \psi_j \rangle$ is the norm of $V^{\pi_j}$'s projection in $\mathrm{span}(V^{\pi_1}, \ldots, V^{\pi_{j-1}})^\perp$ and $u_j$ belongs to $\mathrm{span}(V^{\pi_1}, \ldots, V^{\pi_{j-1}})^\perp$.

By taking absolute value on both sides of Eq (17), we have

$$\epsilon_i |\langle \widehat{w}, \psi_i \rangle| = \left| \widehat{w} \cdot \widehat{A}_i^\| - \sum_{j=d_\delta+1}^{i-1} \langle V^{\pi_i}, \psi_j \rangle \langle \widehat{w}, \psi_j \rangle \right| \leq \left| \widehat{w} \cdot \widehat{A}_i^\| \right| + \sqrt{k} \sum_{j=d_\delta+1}^{i-1} \epsilon_j |\langle \widehat{w}, \psi_j \rangle| . \tag{19}$$

We can now bound $\left| \widehat{w} \cdot \widehat{A}_i^\| \right|$ as follows.

$$\left| \widehat{w} \cdot \widehat{A}_i^\| \right| = \left| \widehat{w}^{(\delta)} \cdot \widehat{A}_i^\| \right| \tag{20}$$

$$= \left| \widehat{w}^{(\delta)} \cdot \left( \widehat{A}_i - \sum_{j=d_\delta+1}^{d} \left\langle \widehat{A}_i, \psi_j \right\rangle \psi_j \right) \right| = \left| \widehat{w}^{(\delta)} \cdot \widehat{A}_i \right| \tag{21}$$

$$\leq \left| \widehat{w}^{(\delta)} \cdot A_i \right| + \left| \widehat{w}^{(\delta)} \cdot (\widehat{A}_i - A_i) \right|$$

$$\leq |w' \cdot A_i| + \left| (\widehat{w}^{(\delta)} - w') \cdot A_i \right| + \left| \widehat{w}^{(\delta)} \cdot (\widehat{A}_i - A_i) \right|$$

$$\leq 0 + (C_\alpha + 1) \sup_\pi \left| (\widehat{w}^{(\delta)} - w') \cdot V^\pi \right| + C_V \left\| \widehat{w}^{(\delta)} \right\|_2 \epsilon_\alpha$$

$$\leq C' C_\alpha \epsilon^{(\delta)} ,$$

for some constant $C' > 0$.

Eq (20) holds because $\widehat{A}^{(\delta)} \widehat{w} = \widehat{A}^{(\delta)} \widehat{w}^{(\delta)} = \mathbf{e}_1$ and $\widehat{A}_i^\|$ belongs to $\mathrm{span}(\widehat{A}^{(\delta)})$. Eq (21) holds because $\widehat{w}^{(\delta)}$ is the minimum norm solution to $\widehat{A}^{(\delta)} x = \mathbf{e}_1$, which implies that $\widehat{w}^{(\delta)} \cdot \psi_i = 0$. The last inequality follows by applying Lemma 2.

We will bound $\epsilon_i |\langle \widehat{w}, \psi_i \rangle|$ by induction on $i = d_\delta + 1, \ldots, d$.

In the base case of $i = d_\delta + 1$,

$$\epsilon_{d_\delta+1} |\langle \widehat{w}, \psi_{d_\delta+1} \rangle| \leq \left| \widehat{w} \cdot \widehat{A}_{d_\delta+1}^\| \right| \leq C' C_\alpha \epsilon^{(\delta)} .$$

Then, by induction through Eq (19), we have for $i = d_\delta + 2, \ldots, d$,

$$\epsilon_i |\langle \widehat{w}, \psi_i \rangle| \leq (\sqrt{k} + 1)^{i-d_\delta-1} C' C_\alpha \epsilon^{(\delta)} .$$

Similar to the deviation of Eq (18), we pick an orthonormal basis $\rho_1, \ldots, \rho_{k+1-i}$ of $\mathrm{span}(V^{\pi_1}, \ldots, V^{\pi_{i-1}})^\perp$ at line 8 of Algorithm 1, then we have that, for any policy $\pi$,

$$|\langle V^\pi, \psi_i \rangle| \leq \sqrt{k} \max_{l \in [k+1-i]} |\langle V^\pi, \rho_l \rangle| \leq \sqrt{k} |\langle V^{\pi_i}, u_i \rangle| \leq \sqrt{k} |\langle V^{\pi_i}, \psi_i \rangle| = \sqrt{k} \epsilon_i .$$

Then we have that term (b) is bounded by

$$(b) = \left| \sum_{i=d_\delta+1}^{d} \langle V^\pi, \psi_i \rangle \langle \widehat{w}, \psi_i \rangle \right| \leq \sum_{i=d_\delta+1}^{d} |\langle V^\pi, \psi_i \rangle| \cdot |\langle \widehat{w}, \psi_i \rangle|$$

$$\leq \sqrt{k} \sum_{i=d_\delta+1}^{d} \epsilon_i \left|\langle \widehat{w}, \psi_i \rangle\right| \leq (\sqrt{k}+1)^{d-d_\delta} C' C_\alpha \epsilon^{(\delta)}.$$

Hence we have that for any policy $\pi$,

$$\left|\langle \widehat{w}, V^\pi \rangle - \left\langle \widehat{w}^{(\delta)}, V^\pi \right\rangle\right| \leq (\sqrt{k}+1)^{d-d_\delta} C' C_\alpha \epsilon^{(\delta)} + \frac{3}{2}\sqrt{k}\delta \left\|w'\right\|_2.$$

$\square$

## H  Proof of Theorem 2

**Theorem 2.** *Consider the algorithm of computing $\widehat{A}$ and any solution $\widehat{w}^{(\delta)}$ to $\widehat{A}^{(\delta)}x = \mathbf{e}_1$ for $\delta = k^{\frac{5}{3}}\epsilon^{\frac{1}{3}}$ and outputting the policy $\pi^{\widehat{w}^{(\delta)}} = \arg\max_{\pi\in\Pi}\left\langle \widehat{w}^{(\delta)}, V^\pi \right\rangle$. Then the policy $\pi^{\widehat{w}^{(\delta)}}$ satisfies that $v^* - \left\langle w^*, V^{\pi^{\widehat{w}^{(\delta)}}} \right\rangle \leq \mathcal{O}\left(k^{\frac{13}{6}}\epsilon^{\frac{1}{3}}\right).$*

*Proof of Theorem 2.* As shown in Lemma 1, we set $C_\alpha = 2k$ and have $\epsilon_\alpha = \frac{4k^2\epsilon}{v^*}$. We have $\|w'\| = \frac{\|w^*\|}{\langle w^*, V^{\pi_1}\rangle}$ and showed that $\langle w^*, V^{\pi_1}\rangle \geq \frac{v^*}{2k}$ in the proof of Lemma 1.

By applying Lemma 2 and setting $\delta = \left(\frac{C_V^4 k^5 \|w^*\|_2 \epsilon}{v^{*2}}\right)^{\frac{1}{3}}$, we have

$$v^* - \left\langle w^*, V^{\pi^{\widehat{w}^{(\delta)}}} \right\rangle = \langle w^*, V^{\pi_1}\rangle \left(\left\langle w', V^{\pi^*} \right\rangle - \left\langle w', V^{\pi^{\widehat{w}^{(\delta)}}} \right\rangle\right)$$

$$\leq \langle w^*, V^{\pi_1}\rangle \left(\left\langle \widehat{w}^{(\delta)}, V^{\pi^*} \right\rangle - \left\langle \widehat{w}^{(\delta)}, V^{\pi^{\widehat{w}^{(\delta)}}} \right\rangle + \mathcal{O}(\sqrt{k}\left\|w'\right\|_2 (\frac{C_V^4 k^5 \|w^*\|_2 \epsilon}{v^{*2}})^{\frac{1}{3}})\right)$$

$$= \mathcal{O}(\sqrt{k}\left\|w^*\right\|_2 (\frac{C_V^4 k^5 \|w^*\|_2 \epsilon}{v^{*2}})^{\frac{1}{3}}).$$

$\square$

## I  Dependency on $\epsilon$

In this section, we would like to discuss a potential way of improving the dependency on $\epsilon$ in Theorems 1 and 2.

Consider a toy example where the returned three basis policies are $\pi_1$ with $V^{\pi_1} = (1, 0, 0)$, $\pi_2$ with $V^{\pi_2} = (1, 1, 1)$ and $\pi_3$ with $V^{\pi_3} = (1, \eta, -\eta)$ for some $\eta > 0$ and $w^* = (1, w_2, w_3)$ for some $w_2, w_3$.

The estimated ratio $\widehat{\alpha}_1$ lies in $[1 + w_2 + w_3 - \epsilon, 1 + w_2 + w_3 + \epsilon]$, and $\widehat{\alpha}_2$ lies in $[1 + \eta w_2 - \eta w_3 - \epsilon, 1 + \eta w_2 - \eta w_3 + \epsilon]$. Suppose that $\widehat{\alpha}_1 = 1 + w_2 + w_3 + \epsilon$ and $\widehat{\alpha}_2 = 1 + \eta w_2 - \eta w_3 + \epsilon$.

By solving

$$\begin{pmatrix} 1 & 0 & 0 \\ w_2 + w_3 + \epsilon & -1 & -1 \\ \eta w_2 - \eta w_3 + \epsilon & -\eta & \eta \end{pmatrix} \widehat{w} = \begin{pmatrix} 1 \\ 0 \\ 0 \end{pmatrix}$$

we can derive $\widehat{w}_2 = w_2 + \frac{\epsilon}{2}(1 + \frac{1}{\eta})$ and $\widehat{w}_3 = w_3 + \frac{\epsilon}{2}(1 - \frac{1}{\eta})$.

The quantity measuring sub-optimality we care about is $\sup_\pi |\langle \widehat{w}, V^\pi \rangle - \langle w^*, V^\pi \rangle|$, which is upper bounded by $C_V \|\widehat{w} - w^*\|_2$. But the $\ell_2$ distance between $\widehat{w}$ and $w^*$ depends on the condition number of $\widehat{A}$, which is large when $\eta$ is small. To obtain a non-vacuous upper bound in Section 3, we introduce another estimate $\widehat{w}^{(\delta)}$ based on the truncated version of $\widehat{A}$ and then upper bound $\left\|\widehat{w}^{(\delta)} - w^*\right\|_2$ and $\sup_\pi \left|\langle \widehat{w}, V^\pi \rangle - \langle \widehat{w}^{(\delta)}, V^\pi \rangle\right|$ separately.

However, it is unclear if $\sup_\pi |\langle \widehat{w}, V^\pi \rangle - \langle w^*, V^\pi \rangle|$ depends on the condition number of $\widehat{A}$. Due to the construction of Algorithm 1, we can obtain some extra information about the set of all policy values.

First, since we find $\pi_2$ before $\pi_3$, $\eta$ must be no greater than 1. According to the algorithm, $V^{\pi_2}$ is the optimal policy when the preference vector is $u_2$ (see line 11 of Algorithm 1 for the definition of $u_2$) and $V^{\pi_3}$ is the optimal policy when the preference vector is $u_3 = (0, 1, -1)$. Note that the angle between $u_2$ and $V^{\pi_2}$ is no greater than 45 degrees according to the definition of $u_2$. Then the values of all policies can only lie in the small box $B = \{x \in \mathbb{R}^3 | |u_2^\top x| \le |\langle u_2, V^{\pi_2} \rangle|, |u_3^\top x| \le |\langle u_3, V^{\pi_3} \rangle|\}$. It is direct to check that for any $x \in B$, $|\langle \widehat{w}, x \rangle - \langle w^*, x \rangle| < (1 + \sqrt{2})\epsilon$. This example illustrates that even when the condition number of $\widehat{A}$ is large, $\sup_\pi |\langle \widehat{w}, V^\pi \rangle - \langle w^*, V^\pi \rangle|$ can be small. It is unclear if this holds in general. Applying this additional information to upper bound $\sup_\pi |\langle \widehat{w}, V^\pi \rangle - \langle w^*, V^\pi \rangle|$ directly instead of through bounding $C_V \|\widehat{w} - w^*\|_2$ is a possible way of improving the term $\epsilon^{\frac{1}{3}}$.

## J    Description of C4 and Proof of Lemma 4

---

**Algorithm 4** C4: Compress Convex Combination using Carathéodory's theorem

---

1: **input** a set of $k$-dimensional vectors $M \subset \mathbb{R}^k$ and a distribution $p \in \mathrm{Simplex}^M$
2: **while** $|M| > k + 1$ **do**
3:     arbitrarily pick $k + 2$ vectors $\mu_1, \ldots, \mu_{k+2}$ from $M$
4:     solve for $x \in \mathbb{R}^{k+2}$ s.t. $\sum_{i=1}^{k+2} x_i(\mu_i \circ 1) = \mathbf{0}$, where $\mu \circ 1$ denote the vector of appending 1 to $\mu$
5:     $i_0 \leftarrow \arg\max_{i \in [k+2]} \frac{|x_i|}{p(\mu_i)}$
6:     **if** $x_{i_0} < 0$ **then** $x \leftarrow -x$
7:     $\gamma \leftarrow \frac{p(\mu_{i_0})}{x_{i_0}}$ and $\forall i \in [k+2], p(\mu_i) \leftarrow p(\mu_i) - \gamma x_i$
8:     remove $\mu_i$ with $p(\mu_i) = 0$ from $M$
9: **end while**
10: **output** $M$ and $p$

---

**Lemma 4.** *Given a set of $k$-dimensional vectors $M \subset \mathbb{R}^k$ and a distribution $p$ over $M$, C4$(M, p)$ outputs $M' \subset M$ with $|M'| \le k + 1$ and a distribution $q \in \mathrm{Simplex}^{M'}$ satisfying that $\mathbb{E}_{\mu \sim q}[\mu] = \mathbb{E}_{\mu \sim p}[\mu]$ in time $\mathcal{O}(|M| k^3)$.*

*Proof.* The proof is similar to the proof of Carathéodory's theorem. Given the vectors $\mu_1, \ldots, \mu_{k+2}$ picked in line 3 of Algorithm 4 and their probability masses $p(\mu_i)$, we solve $x \in \mathbb{R}^{k+2}$ s.t. $\sum_{i=1}^{k+2} x_i(\mu_i \circ 1) = \mathbf{0}$ in the algorithm.

Note that there exists a non-zero solution of $x$ because $\{\mu_i \circ 1 | i \in [k+2]\}$ are linearly dependent. Besides, $x$ satisfies $\sum_{i=1}^{d+2} x_i = 0$.

Therefore,

$$\sum_{i=1}^{d+2} (p(\mu_i) - \gamma x_i) = \sum_{i=1}^{d+2} p(\mu_i).$$

For all $i$, if $x_i < 0$, $p(\mu_i) - \gamma x_i \ge 0$ as $\gamma > 0$; if $x_i > 0$, then $\frac{x_i}{p(\mu_i)} \le \frac{x_{i_0}}{p(\mu_{i_0})} = \frac{1}{\gamma}$ and thus $p(\mu_i) - \gamma x_i \ge 0$.

Hence, after one iteration, the updated $p$ is still a probability over $M$ (i.e., $p(\mu) \ge 0$ for all $\mu \in M$ and $\sum_{\mu \in M} p(M) = 1$). Besides, $\sum_{i=1}^{d+2} (p(\mu_i) - \gamma x_i)\mu_i = \sum_{i=1}^{d+2} p(\mu_i)\mu_i - \gamma \sum_{i=1}^{d+2} x_i\mu_i = \sum_{i=1}^{d+2} p(\mu_i)\mu_i$.

Therefore, after one iteration, the expected value $\mathbb{E}_{\mu \sim p}[\mu]$ is unchanged.

When we finally output $(M', q)$, we have that $q$ is a distribution over $M$ and that $\mathbb{E}_{\mu \sim q}[\mu] = \mathbb{E}_{\mu \sim p}[\mu]$.

Due to line 6 of the algorithm, we know that $x_{i_0} > 0$. Hence $p(\mu_{i_0}) - \gamma x_{i_0} = p(\mu_{i_0}) - \frac{p(\mu_{i_0})}{x_{i_0}} x_{i_0} = 0$.

We remove at least one vector $\mu_{i_0}$ from $M$ and we will run for at most $|M|$ iterations.

Finally, solving $x$ takes $\mathcal{O}(k^3)$ time and thus, Algorithm 4 takes $\mathcal{O}(|M| \, k^3)$ time in total.  $\square$

## K   Flow Decomposition Based Approach

We first introduce an algorithm based on the idea of flow decomposition.

For that, we construct a layer graph $G = ((L^{(0)} \cup \ldots \cup L^{(H+1)}), E)$ with $H + 2$ pairwise disjoint layers $L^{(0)}, \ldots, L^{(H+1)}$, where every layer $t \leq H$ contains a set of vertices labeled by the (possibly duplicated) states reachable at the corresponding time step $t$, i.e., $\{s \in \mathcal{S} | \Pr(S_t = s | S_0 = s_0) > 0\}$.

Let us denote by $x_s^{(t)}$ the vertex in $L^{(t)}$ labeled by state $s$.

Layer $L^{(H+1)} = \{x_*^{(H+1)}\}$ contains only an artificial vertex, $x_*^{(H+1)}$, labeled by an artificial state $*$.

For $t = 0, \ldots, H - 1$, for every $x_s^{(t)} \in L^{(t)}$, $x_{s'}^{(t+1)} \in L^{(t+1)}$, we connect $x_s^{(t)}$ and $x_{s'}^{(t+1)}$ by an edge labeled by $(s, s')$ if $P(s'|s, \pi(s)) > 0$. Every vertex $x_s^{(H)}$ in layer $H$ is connected to $x_*^{(H+1)}$ by one edge, which is labeled by $(s, *)$.

We denote by $E^{(t)}$ the edges between $L^{(t)}$ and $L^{(t+1)}$. Note that every trajectory $\tau = (s_0, s_1, \ldots, s_H)$ corresponds to a single path $(x_{s_0}^{(0)}, x_{s_1}^{(1)}, \ldots, x_{s_H}^{(H)}, x_*^{(H+1)})$ of length $H + 2$ from $x_{s_0}^{(0)}$ to $x_*^{(H+1)}$.

This is a one-to-one mapping and in the following, we use path and trajectory interchangeably.

The policy corresponds to a $(x_{s_0}^{(0)}, x_*^{(H+1)})$-flow with flow value 1 in the graph $G$. In particular, the flow is defined as follows.

When the layer $t$ is clear from the context, we actually refer to vertex $x_s^{(t)}$ by saying vertex $s$.

For $t = 0, \ldots, H - 1$, for any edge $(s, s') \in E^{(t)}$, let $f : E \to \mathbb{R}^+$ be defined as

$$f(s, s') = \sum_{\tau:(s_t^\tau, s_{t+1}^\tau)=(s,s')} q^\pi(\tau), \tag{22}$$

where $q^\pi(\tau)$ is the probability of $\tau$ being sampled.

For any edge $(s, *) \in E^{(H)}$, let $f(s, *) = \sum_{(s',s) \in E^{(H-1)}} f(s', s)$. It is direct to check that the function $f$ is a well-defined flow. We can therefore compute $f$ by dynamic programming.

For all $(s_0, s) \in E^{(0)}$, we have $f(s_0, s) = P(s|s_0, \pi(s_0))$ and for $(s, s') \in E^{(t)}$,

$$f(s, s') = P(s'|s, \pi(s)) \sum_{s'':(s'',s) \in E^{(t-1)}} f(s'', s). \tag{23}$$

Now we are ready to present our algorithm by decomposing $f$ in Algorithm 5.

Each iteration in Algorithm 5 will zero out at least one edge and thus, the algorithm will stop within $|E|$ rounds.

---
**Algorithm 5** Flow decomposition based approach
---
1: initialize $Q \leftarrow \emptyset$.
2: calculate $f(e)$ for all edge $e \in E$ by dynamic programming according to Eq (23)
3: **while** $\exists e \in E$ s.t. $f(e) > 0$ **do**
4:   pick a path $\tau = (s_0, s_1, \ldots, s_H, *) \in L^{(0)} \times L^{(1)} \times \ldots \times L^{(H+1)}$ s.t. $f(s_i, s_{i+1}) > 0 \ \forall i \geq 0$

5:   $f_\tau \leftarrow \min_{e \text{ in } \tau} f(e)$
6:   $Q \leftarrow Q \cup \{(\tau, f_\tau)\}$, $f(e) \leftarrow f(e) - f_\tau$ for $e$ in $\tau$
7: **end while**
8: output $Q$
---

**Theorem 4.** *Algorithm 5 outputs $Q$ satisfying that $\sum_{(\tau, f_\tau) \in Q} f_\tau \Phi(\tau) = V^\pi$ in time $\mathcal{O}(H^2 |\mathcal{S}|^2)$.*

The core idea of the proof is that for any edge $(s, s') \in E^{(t)}$, the flow on $(s, s')$ captures the probability of $S_t = s \wedge S_{t+1} = s'$ and thus, the value of the policy $V^\pi$ is linear in $\{f(e) | e \in E\}$.

The output $Q$ has at most $|E|$ number of weighted paths (trajectories). We can further compress the representation through C4, which takes $O(|Q| k^3)$ time.

**Corollary 2.** *Executing Algorithm 5 with the output $Q$ first and then running C4 over $\{(\Phi(\tau), f_\tau) | (\tau, f_\tau) \in Q\}$ returns a $(k + 1)$-sized weighted trajectory representation in time $\mathcal{O}(H^2 |\mathcal{S}|^2 + k^3 H |\mathcal{S}|^2)$.*

We remark that the running time of this flow decomposition approach underperforms that of the expanding and compressing approach (see Theorem 3) whenever $|\mathcal{S}|H + |\mathcal{S}|k^3 = \omega(k^4 + k|\mathcal{S}|)$.

## L  Proof of Theorem 4

**Theorem 4.** *Algorithm 5 outputs $Q$ satisfying that $\sum_{(\tau, f_\tau) \in Q} f_\tau \Phi(\tau) = V^\pi$ in time $\mathcal{O}(H^2 |\mathcal{S}|^2)$.*

*Proof.* **Correctness:**  The function $f$ defined by Eq (22) is a well-defined flow since for all $t = 1, \ldots, H$, for all $s \in L^{(t)}$, we have that

$$\sum_{s' \in L^{(t+1)}:(s,s') \in E^{(t)}} f(s, s') = \sum_{s' \in L^{(t+1)}:(s,s') \in E^{(t)}} \sum_{\tau:(s_t^\tau, s_{t+1}^\tau)=(s,s')} q^\pi(\tau) = \sum_{\tau:s_t^\tau=s} q^\pi(\tau)$$

$$= \sum_{s'' \in L^{(t-1)}:(s'',s) \in E^{(t-1)}} f(s'', s).$$

In the following, we first show that Algorithm 5 will terminate with $f(e) = 0$ for all $e \in E$.

First, after each iteration, $f$ is still a feasible $(x^{(0)}, x^{(H+1)})$-flow feasible flow with the total flow out-of $x^{(0)}$ reduced by $f_\tau$. Besides, for edge $e$ with $f(e) > 0$ at the beginning, we have $f(e) \geq 0$ throughout the algorithm because we never reduce $f(e)$ by an amount greater than $f(e)$.

Then, since $f$ is a $(x^{(0)}, x^{(H+1)})$-flow and $f(e) \geq 0$ for all $e \in E$, we can always find a path $\tau$ in line 4 of Algorithm 5.

Otherwise, the set of vertices reachable from $x^{(0)}$ through edges with positive flow does not contain $x^{(H+1)}$ and the flow out of this set equals the total flow out-of $x^{(0)}$. But since other vertices are not reachable, there is no flow out of this set, which is a contradiction.

In line 6, there exists at least one edge $e$ such that $f(e) > 0$ is reduced to 0. Hence, the algorithm will run for at most $|E|$ iterations and terminate with $f(e) = 0$ for all $e \in E$.

Thus we have that for any $(s, s') \in E^{(t)}$, $f(s, s') = \sum_{(\tau, f_\tau) \in Q:(s_t^\tau, s_{t+1}^\tau)=(s,s')} f_\tau$.

Then we have

$$V^\pi = \sum_\tau q^\pi(\tau)\Phi(\tau) = \sum_\tau q^\pi(\tau)\left(\sum_{t=0}^{H-1} R(s_t^\tau, \pi(s_t^\tau))\right)$$

$$= \sum_{t=0}^{H-1} \sum_\tau q^\pi(\tau) R(s_t^\tau, \pi(s_t^\tau))$$

$$= \sum_{t=0}^{H-1} \sum_{(s,s') \in E^{(t)}} R(s, \pi(s)) \sum_{\tau:(s_t^\tau, s_{t+1}^\tau)=(s,s')} q^\pi(\tau)$$

$$= \sum_{t=0}^{H-1} \sum_{(s,s') \in E^{(t)}} R(s, \pi(s)) f(s, s')$$

$$= \sum_{t=0}^{H-1} \sum_{(s,s') \in E^{(t)}} R(s, \pi(s)) \sum_{(\tau, f_\tau) \in Q:(s_t^\tau, s_{t+1}^\tau)=(s,s')} f_\tau$$

$$= \sum_{(\tau, f_\tau) \in Q} f_\tau \left( \sum_{t=0}^{H-1} R(s_t^\tau, \pi(s_t^\tau)) \right)$$

$$= \sum_{(\tau, f_\tau) \in Q} f_\tau \Phi(\tau).$$

**Computational complexity:** Solving $f$ takes $\mathcal{O}(|E|)$ time. The algorithm will run for $\mathcal{O}(|E|)$ iterations and each iteration takes $\mathcal{O}(H)$ time. Since $|E| = \mathcal{O}(|\mathcal{S}|^2 H)$, the total running time of Algorithm 5 is $\mathcal{O}(|\mathcal{S}|^2 H^2)$. C4 will take $\mathcal{O}(k^3 |E|)$ time. $\qquad\square$

## M   Proof of Theorem 3

**Theorem 3.** *Algorithm 2 outputs $F^{(H)}$ and $\beta^{(H)}$ satisfying that $\left|F^{(H)}\right| \leq k+1$ and $\sum_{\tau \in F^{(H)}} \beta^{(H)}(\tau) \Phi(\tau) = V^\pi$ in time $\mathcal{O}(k^4 H |\mathcal{S}| + kH |\mathcal{S}|^2)$.*

*Proof.* **Correctness:** C4 guarantees that $\left|F^{(H)}\right| \leq k+1$.

We will prove $\sum_{\tau \in F^{(H)}} \beta_\tau^{(H)} \Phi(\tau) = V^\pi$ by induction on $t = 1, \ldots, H$.

Recall that for any trajectory $\tau$ of length $h$, $J(\tau) = \Phi(\tau) + V(s_h^\tau, H - h)$ was defined as the expected return of trajectories (of length $H$) with the prefix being $\tau$.

In addition, recall that $J_{F^{(t)}} = \{J(\tau \circ s) | \tau \in F^{(t)}, s \in \mathcal{S}\}$ and $p_{F^{(t)}, \beta^{(t)}}$ was defined by letting $p_{F^{(t)}, \beta^{(t)}}(\tau \circ s) = \beta^{(t)}(\tau) P(s | s_t^\tau, \pi(s_t^\tau))$.

For the base case, we have that at $t = 1$

$$V^\pi = R(s_0, \pi(s_0)) + \sum_{s \in \mathcal{S}} P(s | s_0, \pi(s_0)) V^\pi(s, H-1) = \sum_{s \in \mathcal{S}} P(s | s_0, \pi(s_0)) J((s_0) \circ s)$$

$$= \sum_{s \in \mathcal{S}} p_{F^{(0)}, \beta^{(0)}}((s_0) \circ s) J((s_0) \circ s) = \sum_{\tau \in F^{(1)}} \beta^{(1)}(\tau) J(\tau).$$

Suppose that $V^\pi = \sum_{\tau' \in F^{(t)}} \beta^{(t)}(\tau') J(\tau')$ holds at time $t$, then we prove the statement holds at time $t + 1$.

$$V^\pi = \sum_{\tau' \in F^{(t)}} \beta^{(t)}(\tau') J(\tau') = \sum_{\tau' \in F^{(t)}} \beta^{(t)}(\tau')(\Phi(\tau') + V^\pi(s_t^{\tau'}, H - t))$$

$$= \sum_{\tau' \in F^{(t)}} \beta^{(t)}(\tau') \left( \Phi(\tau') + \sum_{s \in \mathcal{S}} P(s | s_t^{\tau'}, \pi(s_t^{\tau'})) \left( R(s_t^{\tau'}, \pi(s_t^{\tau'})) + V^\pi(s, H - t) \right) \right)$$

$$= \sum_{\tau' \in F^{(t)}} \sum_{s \in \mathcal{S}} \beta^{(t)}(\tau') P(s | s_t^{\tau'}, \pi(s_t^{\tau'})) \left( \Phi(\tau') + R(s_t^{\tau'}, \pi(s_t^{\tau'})) + V^\pi(s, H - t) \right)$$

$$= \sum_{\tau' \in F^{(t)}} \sum_{s \in \mathcal{S}} \beta^{(t)}(\tau') P(s | s_t^{\tau'}, \pi(s_t^{\tau'})) (\Phi(\tau' \circ s) + V^\pi(s, H - t))$$

$$= \sum_{\tau' \in F^{(t)}} \sum_{s \in \mathcal{S}} p_{F^{(t)}, \beta^{(t)}}(\tau' \circ s) J(\tau' \circ s)$$

$$= \sum_{\tau \in F^{(t+1)}} \beta^{(t+1)}(\tau) J(\tau).$$

By induction, the statement holds at $t = H$ by induction, i.e., $V^\pi = \sum_{\tau \in F^{(H)}} \beta_\tau^{(H)} J(\tau) = \sum_{\tau \in F^{(H)}} \beta_\tau^{(H)} \Phi(\tau)$.

**Computational complexity:** Solving $V^\pi(s, h)$ for all $s \in \mathcal{S}, h \in [H]$ takes time $\mathcal{O}(kH |\mathcal{S}|^2)$. In each round, we need to call C4 for $\leq (k+1) |S|$ vectors, which takes $\mathcal{O}(k^4 |\mathcal{S}|)$ time. Thus, we need $\mathcal{O}(k^4 H |\mathcal{S}| + kH |\mathcal{S}|^2)$ time in total. $\qquad\square$

# N Example of maximizing individual objective

**Observation 2.** *Assume there exist $k > 2$ policies that together assemble $k$ linear independent value vectors. Consider the $k$ different policies $\pi_1^*, \ldots, \pi_k^*$ that each $\pi_i^*$ maximizes the objective $i \in [k]$. Then, their respective value vectors $V_1^*, \ldots, V_k^*$ are not necessarily linearly independent. Moreover, if $V_1^*, \ldots, V_k^*$ are linearly depended it does not mean that $k$ linearly independent value vectors do not exists.*

*Proof.* For simplicity, we show an example with a horizon of $H = 1$ but the results could be extended to any $H \geq 1$. We will show an example where there are $4$ different value vectors, where $3$ of them are obtained by the $k = 3$ policies that maximize the $3$ objectives and have linear dependence.

Consider an MDP with a single state (also known as Multi-arm Bandit) with $4$ actions with deterministic reward vectors (which are also the expected values of the $4$ possible policies in this case):

$$r(1) = \begin{pmatrix} 8 \\ 4 \\ 2 \end{pmatrix}, \quad r(2) = \begin{pmatrix} 1 \\ 2 \\ 3 \end{pmatrix}, \quad r(3) = \begin{pmatrix} 85/12 \\ 25/6 \\ 35/12 \end{pmatrix} \approx \begin{pmatrix} 7.083 \\ 4.167 \\ 2.9167 \end{pmatrix}, \quad r(4) = \begin{pmatrix} 1 \\ 3 \\ 2 \end{pmatrix}.$$

Denote $\pi^a$ as the fixed policy that always selects action $a$. Clearly, policy $\pi^1$ maximizes the first objective, policy $\pi^2$ the third, and policy $\pi^3$ the second ($\pi^4$ do not maximize any objective). However,

- $r(3)$ linearly depends on $r(1)$ and $r(2)$ as

$$\frac{5}{6}r(1) + \frac{5}{12}r(2) = r(3).$$

- In addition, $r(4)$ is linearly independent in $r(1), r(2)$: Assume not. Then, there exists $\beta_1, \beta_2 \in \mathbb{R}$ s.t.:

$$\beta_1 \cdot r(1) + \beta_2 \cdot r(2) = \begin{pmatrix} 8\beta_1 + \beta_2 \\ 4\beta_1 + 2\beta_2 \\ 2\beta_1 + 3\beta_2 \end{pmatrix} = \begin{pmatrix} 1 \\ 3 \\ 2 \end{pmatrix} = r(4).$$

Hence, the first equations imply $\beta_2 = 1 - 8\beta_1$, and $4\beta_1 + 2 - 16\beta_1 = 3$, hence $\beta_1 = -\frac{1}{12}$ and $\beta_2 = \frac{5}{3}$. Assigning in the third equation yields $-\frac{1}{6} + 5 = 2$ which is a contradiction.

$\square$

