# OpenReview forum: "Eliciting User Preferences for Personalized Multi-Objective Decision Making through Comparative Feedback"
_NeurIPS.cc/2023/Conference — NeurIPS 2023 poster_

### Official Review · Reviewer_BDkG · 2023-07-04

**Soundness:** 2 fair
**Presentation:** 1 poor
**Contribution:** 2 fair
**Rating:** 3
**Confidence:** 4

**Summary:**

The paper presents a multi-objective decision making framework that accommodates different user preferences over multiple objectives. Furthermore, the framework facilitates learning these preferences via policy comparisons. The underlying model is assumed to be a finite horizon Markov decision process, while the user preference relation is assumed to be a linear combination of objective values. The preference relation is subsequently learned using user feedback taking two forms: policy comparison, and the comparison of two small sets of representative trajectories, respectively. The paper then suggests an algorithm for finding near-optimal policies using a number of comparison queries that scales quasi-linearly with the number of objectives.


**Strengths:**

The paper addresses an important problem in decision making under uncertainty with multiple objectives which has a wide range of potential applications.


**Weaknesses:**

However, the paper has several shortcomings. First, the presentation lacks a small running example and therefore it is quite difficult to follow the technical details of the proposed approach. Second, I was surprised by the abrupt ending of the paper (with Theorem 3) without any follow-up or discussion. It gives the impression that the submission was rushed right before the deadline which is kinda disappointing. Third, the paper lacks any empirical evidence of the proposed approach and therefore we don't have any idea whether the approach is practical or not. For example, if the MDP has a very complex state (and/or action) space, how would one best represent and present the trajectories to the user to elicit feedback. While it seems straightforward in a toy-like setting, I don't think the preference elicitation scheme based on trajectories is actually practical in a real setting.


**Questions:**

No specific questions.


**Limitations:**

I found the paper closely related to previous work by (Marinescu et al, 2017) which also looks at multi-objective decision making in the case of influence diagram models (which are actually more general than finite-horizon MDPs) and an unknown linear preference relation between objectives. The goal however, is slightly different, in that it is concerned with finding possibly optimal policies (which are optimal under at least one instantiation of the preference vector).

(Marinescu et al, 2017) Marinescu, R., Razak, A. and Wilson, N. 'Multi-Objective Influence Diagrams with Possibly Optimal Policies', Proceedings of the Thirty-First AAAI Conference on Artificial Intelligence (AAAI-17), San Francisco, California, USA, 4 – 9 February, pp. 3783 - 3789.

---

> ### Author Rebuttal · Authors · 2023-08-10
>
> Thank you for the informative review. We will address some of your comments below:
>
> > A small running example.
>
> We will extend our usage of the Google maps example (lines 16, 56, 300) through the paper to improve the paper's readability. Thank you for your suggestion.
>
>
> > On abrupt ending of the paper.
>
> We have addressed this in the general response. Please refer to the general response for details.
>
> > Third, the paper lacks any empirical evidence of the proposed approach and therefore we don't have any idea whether the approach is practical or not. For example, if the MDP has a very complex state (and/or action) space, how would one best represent and present the trajectories to the user to elicit feedback. While it seems straightforward in a toy-like setting, I don't think the preference elicitation scheme based on trajectories is actually practical in a real setting.
>
> We agree that in cases where the state representation is complex, the task is challenging, but this challenge is a result of either the complexity of the system or a non-efficient design of the MDP that describes the environment (and it is not due to our solution). Preference elicitation scheme based on trajectories is actually quite practical. For example, openAI using them (where each trajectory is a chatGPT response). We will address this in the final version of the paper!
>
> > On the reference (Marinescu et al, 2017)
>
> Thank you for the reference. While we were not familiar with this work, in both the Introduction (lines 21- “While prior works have concentrated on approximating the Pareto-optimal solution set”, 81- “ Prior research has explored various approaches, such as assuming linear preferences or Bayesian Settings, or finding an approximated Pareto frontier”) and in the (additional) related work section (App B), we did describe some works with similar approaches  to that of (Marinescu et al, 2017). Namely, the approach of finding policies in the pareto front. As the reviewer said, our goal is different. Instead of finding a possibly optimal policy, we concentrate on specific user preferences and find a policy that is optimal for that specific user. Just like the ideal car for one person could be a Chevrolet Spark (small) and for another, it is a Ford Ranger (a truck). We promise to cite the work of (Marinescu et al, 2017) as well in the final version of the paper!

---

### Official Review · Reviewer_pTmr · 2023-07-05

**Soundness:** 4 excellent
**Presentation:** 2 fair
**Contribution:** 2 fair
**Rating:** 5
**Confidence:** 4

**Summary:**

The paper presents a novel approach to elicit the unknown linear weights used to aggregate multi-objective valuations in sequential decision-making problems. The authors first consider comparative queries where a user can compare policies (instead of trajectories as in previous work). They investigate then the case where a policy can be replaced and approximated by a set of weighted trajectories.

**Strengths:**

The paper investigates a novel approach for preference elicitation in multi-objective problems.

The authors obtain upper-bounds on the number of queries needed to obtain a near optimal policy for a given user.

The approach scales well with the number of users.

**Weaknesses:**

I think the obtained results are mostly of theoretical interest, because the type of considered queries is not very practical, in my opinion. Directly comparing policies is naturally usually not possible for humans, which motivates the second case investigated in this paper. However, even comparing two sets of weighted trajectories would be very hard for humans.

It is not clear if the proposed approach based on geometric and algebraic results can work in presence of noisy feedback from the user, which is a more realistic setting.

The writing, while reasonably clear, seems to be a bit rushed. Notably, the paper ends abruptly, without any conclusion.

**Questions:**

1) In footnote 1, why do the authors mention "nearly"? I think under the assumptions they make, optimal policies for every possible user preferences are in the set of Pareto optimal solutions.

2) Are there situations where comparing sets of weighted trajectories would be practical for humans?

3) Why do you not assume that the weight vector w^* is normalized since dividing by a positive value doesn't change the preferences over policies?

4) Can the proposed technique be adapted if there is noise in the user's comparisons?

Some minor issues:
- line 107: V(\pi) -> V^\pi
- line 128: In line 96, policies are introduced as stochastic, which seems to contradict with the explicit policy representation.
- line 259: vacuous -> non-vacuous?

**Limitations:**

Not applicable, this is a theoretical paper.

---

> ### Author Rebuttal · Authors · 2023-08-10
>
> We thank the reviewer for the informative review. We address reviewer’s questions in the following and will incorporate the suggestions in the final version.
>
> > The type of considered queries is not very practical. Directly comparing policies is naturally usually not possible for humans, which motivates the second case investigated in this paper. However, even comparing two sets of weighted trajectories would be very hard for humans. Are there situations where comparing sets of weighted trajectories would be practical for humans?
>
> > [question #2] “Are there situations where comparing sets of weighted trajectories would be practical for humans?”
>
> Consider a casino with slot machines in which someone puts some money (e.g., a dollar) and receives money with some probability (e.g., $0$ w.p. 91% vs. $10$ w.p, 9%). Describing each game policy using a weighted set can be very practical. The same goes for investment strategy.
> Notice that the weights express probabilities or variance of the outcome.
> Another example is driving. E.g., when the goal is to get from home to the airport, taking a specific route takes 40 minutes 90% of the time, but it can take 3 hours in case of an accident (which happens w.p. 10%) vs. taking the subway which always has a duration of 1 hour.
>
>
> > On noisy feedback model and abrupt ending of the paper.
>
> We have addressed these in the general response. Please refer to the general response for details.
>
> > In footnote 1, why do the authors mention "nearly"?
>
> That is correct, we will definitely remove the word ‘nearly’.
>
> > Why do you not assume that the weight vector $w^*$ is normalized since dividing by a positive value doesn't change the preferences over policies?
>
> In terms of personalized optimal policy, you are correct that normalizing does not change the personalized optimal policy. However, the magnitude affects the “indistinguishability’’ in the feedback model. By not normalizing $w^*$, we allow users to have varying levels of discernment. For instance, consider two policies with values $(\epsilon, 1)$ and $(0,1)$, respectively. For a user with preference vector $w^* = (1,1)$, they cannot differentiate between $(\epsilon, 1)$ and $(0,1)$ and will respond with "indistinguishable" when queried to compare these two policies. On the other hand, for a user with preference vector $w^* = (2,2)$, they can distinguish between $(\epsilon, 1)$ and $(0,1)$ and will indicate a preference for the former policy.
>
> > On minor comments
>
> Thank you for pointing out the typos. For explicit policy representation, it should be corrected from a mapping from $S$ to $A$ to a mapping from $S$ to the simplex over $A$. We will fix the typos in the final version.

---

> > ### Comment · Reviewer_pTmr · 2023-08-16
> > **Rebuttal**
> >
> > Thank you for the rebuttal. Regarding the set of weighted trajectories, in all your examples, the trajectories are summarized by some scalar values. However, in the multiobjective setting, comparisons become much harder in my opinion. That was what I meant.
> >
> > Regarding the discussion at the end of the paper and the addition of a running example, it's not clear how the authors would manage that, given the limited space they already have. For now, since I don't see any updated submission, I will keep my score unchanged.

---

> > > ### Author Response · Authors · 2023-08-17
> > >
> > > Thank you for reading our rebuttal and quick response.
> > >
> > > We will include this discussion in the additional content page that is allowed for accepted papers (https://neurips.cc/Conferences/2023/CallForPapers).
> > >
> > > Regarding the trajectory sets- we apologize for the misunderstanding.
> > > Here are extensions of two of our examples for multiple objectives:
> > > Investment strategy- when considering investing money in a specific portfolio (policy), clear objectives are:
> > >
> > > 1. Expected reward.
> > >
> > > 2. Certain risk measures that are non-linear in the reward (e.g., VaR, CVaR).
> > >
> > > 3. The amount of money that the investor might lose. For example, in short selling, one could lose more money than they initially invested.
> > >
> > > 4. The Eco-friendliness of the portfolio throughout time. For example, investing in big oil companies might be less eco-friendly compared to, e.g., Seventh Generation (a cleaning product company that sells eco-friendly products).
> > >
> > > Notice that while portfolios could have the same expected reward, their other criteria could vary significantly.
> > > One example of the weighted trajectory set of an investment strategy is:
> > >
> > > 1. 90% weight on a trajectory of reward 7, low-risk measures in all states, huge losses in a few states, and very eco-friendly investments;
> > > 2. 10% weight on a trajectory of reward -6, high-risk measures in half of the states and low-risk measures in the other half, huge losses in some states, and with the least eco-friendly investments.
> > >
> > > Commuting to the airport- besides commute time, one could consider potential productivity (as people might be more productive when they are on the train), comfort, safety, and again eco-friendliness.
> > >
> > > We would be happy to incorporate these examples in the paper!

---

### Official Review · Reviewer_5457 · 2023-07-11

**Soundness:** 3 good
**Presentation:** 3 good
**Contribution:** 3 good
**Rating:** 7
**Confidence:** 2

**Summary:**

The paper provides a multi-objective decision-making framework to learn approximate optimal personalized policies by eliciting user preferences via comparative queries. The authors characterize theoretically two feedback models, explicit queries and weights queries sets, and derive sound algorithms with proven upper bounds on the learning complexity.

**Strengths:**

To the best of my knowledge, the proposed algorithms and theoretical analysis are novel and interesting. Eliciting multi-objective user preferences over policies has a tremendous impact on many (real-world) settings in which decision-making systems aid humans in daily tasks. The paper is well-written. All the theoretical assumptions, results and algorithms are well-motivated and clearly described. I checked the proofs (and skimmed through the appendix), but a more rigorous analysis would require more time.

**Weaknesses:**

In my opinion, I can see two main areas of improvement (1) extending the algorithm to noisy feedback from the users and (2) providing some empirical evidence about the effectiveness of the proposed algorithms.

Lastly, the paper is dense with information, but it ends abruptly. I would have enjoyed a paragraph either summarizing the main findings or expanding the current research to consider its potential practical impact (e.g., challenges in implementation).

**Questions:**

- If we consider human-decision making, do you think the two proposed feedback models are realistically usable by humans or do you think they might work well with agents, for example, in cooperative AI settings?

**Limitations:**

The authors adequately address all the limitations.

---

> ### Author Rebuttal · Authors · 2023-08-10
>
> We thank the reviewer for the informative review.
>
> > On noisy feedback model and abrupt ending of the paper.
>
> We have addressed these questions in the general response. Please refer to the general response for details.
>
> > Empirical evidence
>
> This work is purely theoretical and our focus is on theoretical analysis.
>
> > Do you think the two proposed feedback models are realistically usable by humans or do you think they might work well with agents, for example, in cooperative AI settings?
>
> Indeed, openAI trained ChatGPT (and it is still training it) using human feedback regarding comparative queries. As mentioned in the response to Reviewer wfQV’s question on “Examples on how to present the obtained trajectories in an easy format to the user”, in the ChatGPT example (line 50), each trajectory is some text generated by a ChatGPT session with a different history.

---

> > ### Comment · Reviewer_5457 · 2023-08-11
> >
> > I thank the authors for the rebuttal! At the present time, I do not have additional questions, since the various answers cleared most of my doubts. Lastly, I stand with my previous scores.

---

### Official Review · Reviewer_Gnoo · 2023-07-18

**Soundness:** 4 excellent
**Presentation:** 3 good
**Contribution:** 3 good
**Rating:** 7
**Confidence:** 4

**Summary:**

_**Edit:** I have read the authors' rebuttal, and they have addressed all the concerns raised in my review._

The paper considers the problem of multi-objective reinforcement learning, when there is an unknown linear preference over the objectives, and approximate preference comparisons can be performed. The goal is to find an optimal policy (according to the true preference) while minimizing the number of comparisons.

The main task is identifying the unknown preference, which the paper shows how to do using Algorithm 1, Algorithm 3, and Theorem 1. Once the preference is (approximately) identified, Theorem 1 shows that solving the RL problem for this approximate preference returns a policy that is competitive with the optimal policy for the true preference.

Additionally, the paper also considers how preference comparisons could be implemented. One option is to ask the user to compare two policies directly, which might be hard to do. Because of this, the paper considers the case when the user can compare trajectories from two policies, which might be more interpretable to the user. To this end, the paper shows how to efficiently find a small (O(k)) weighted set of trajectories such that their weighted values are equal to the value of the policy (Algorithm 2).

**Strengths:**

- The procedure for identification of linear preference is not limited to just RL and can be used in a wider setting, making it of general interest.
- Algorithm 1 adapts to $d \le k$, which could be helpful in cases when there are many redundant objectives.
- The weighted trajectory set representation of a policy is novel, interpretable, and possibly of wider interest.

**Weaknesses:**

- Line 220: “we only need to run Algorithm 1 once for all users”. This seems to be incorrect, as Algorithm 1 uses preference comparisons which depend on the user. This means the leading factor in the bound on line 256 should change from $(k^2+n)$ to $n (k^2 + 1)$.
- The paper assumes the existence of a “do nothing” action, which might require a modified MDP. While this appears straightforward, it seems to me it requires domain-specific knowledge in order to modify it such that the “do nothing” policy has minimal value (which is the true requirement). E.g. in the chess example, perhaps refusing to play is considered a draw, which is better than a loss. Thus, a “do nothing” action would need to somehow always lose.

**Questions:**

- Proof of Lemma 1: in the first inequality, shouldn't it be possible to reduce $k \epsilon$ to $\epsilon$? This would make Lemma 1 applicable to when $\epsilon \le v^\*/(2k)$ rather than $v^*/(2k^2)$.
- Is it possible to approximately estimate $\epsilon$ by comparing $\pi^1$ against a scaled version of itself (similar to the binary search in Algorithm 3)? If $\hat{\epsilon} \approx \epsilon$ can be estimated this way, can Theorem 2 be used with $\delta = k^{5/3} \hat{\epsilon}^{1/3}$, avoiding the problem of having to guess the hyperparameter $\delta$?
- Have you considered the case when the user can compare policy value vectors? In this case, we can query with “artificial” values directly (e.g. basis vectors), requiring no interaction with the MDP at all. Perhaps this makes the problem too simple?
- The preference identification procedure can be extended to general, non-tabular RL case. Is the same true for efficiently finding weighted trajectory sets?

**Limitations:**

The paper requires that preferences are linear, which is a fairly strong assumption. However, it is a good first step.

---

> ### Author Rebuttal · Authors · 2023-08-10
>
> Thank you for the helpful and detailed review. We especially appreciate the kind comments about the novelty of our main contributions and their broader implications. Your suggestions are very helpful and we will execute them in the camera ready version. We address the reviewer’s questions below:
>
> > On  “we only need to run Algorithm 1 once for all users”.
>
> We thank the reviewer for pointing this out. The reviewer is correct that the sentence ”we only need to run Algorithm 1 once for all users” was imprecise. We intended to say that we don’t have to fully run Algorithm 1 for $n$ times. In fact, when there are $n$ different users, we will run Algorithm 1 at most $k$ times instead of $n$ times. The reason is that Algorithm 1 only utilizes preference comparisons while searching for $\pi_1$ (lines 1-6), and not for $\pi_2, \ldots, \pi_k$ (which contributes to the $k^2$ factor in computational complexity). As there are at most $k$ candidates of $\pi_1$, namely $\pi^{e_1}, \ldots, \pi^{e_k}$, we execute lines 1-6 of Algorithm 1 for $n$ rounds and lines 7-15 for only $k$ rounds. Consequently, we need to update the leading factor of our computational complexity bound from $(k^2+n)$ to $(k^3+n)$.
>
> > “do nothing” action seems to require domain-specific knowledge in order to modify it such that the “do nothing” policy has minimal value (which is the true requirement).
>
> This is correct and we'll be sure to explain this in the final version of the paper, we thank you for that.
>
> > On the proof of Lemma 1: in the first inequality, shouldn't it be possible to reduce $k\epsilon$ to $\epsilon$?
>
> We agree with the reviewer that we are able to reduce $k\epsilon$ to $\epsilon$ if we compare every pair of $\pi^{e_i}$ and $\pi^{e_j}$. However, our prime focus is on minimizing the number of queries and this will increase the number of queries (used for searching $\pi_1$) from $k$ to $k^2$. Hence, it is a trade off between the number of queries and accuracy. We'll be sure to address that in the final version of the paper.
>
> > Is it possible to approximately estimate $\epsilon$ by comparing $\pi_1$ against a scaled version of itself (similar to the binary search in Algorithm 3) to avoid guessing the hyperparameter $\delta$?
>
> This is an interesting observation! We agree with the reviewer that it seems correct to estimate $\epsilon$ in this way. We will add it right after Theorem 2.
>
> > On direct value vector comparisons. Perhaps this makes the problem too simple?
>
> Yes, if that would have been the case the problem would be very simple. However, it does not necessarily align with our motivation, where the goal is to compare realistic policies.
>
> >  Extension of weighted trajectory sets to general, non-tabular RL case.
>
> Our current techniques rely on a finite set of space. Generalizing them to a large environment would require a different technique. It could be a very interesting direction for future work.
>
> > On linear preference assumption.
>
> We want to address that linearity is a standard assumption behind many works in multi-objective decision making and multi-objective RL.

---

> > ### Comment · Reviewer_Gnoo · 2023-08-10
> >
> > Thank you for the reply. All my concerns have been addressed, except for the (very minor) one regarding Lemma 1:
> >
> > > We agree with the reviewer that we are able to reduce $k \epsilon$ to $\epsilon$ if we compare every pair of $\pi^{e_i}$ and $\pi^{e_j}$. However, our prime focus is on minimizing the number of queries and this will increase the number of queries (used for searching $\pi_1$) from $k$ to $k^2$. Hence, it is a trade off between the number of queries and accuracy. We'll be sure to address that in the final version of the paper.
> >
> > I believe the $O(k)$ approach in the paper is enough to get $\epsilon$ optimality. Here is an attempt at a proof, by induction. In the base case, comparing $k = 2$ policies, it is clear the returned policy will be $\epsilon$-optimal. For $k > 2$, first run the comparison on the first $k-1$ policies to get an $\epsilon$-optimal policy $\pi^*$: for all $i \in [k-1]$, $w \pi^* \ge w \pi_i - \epsilon$. Then, compare $\pi_k$ and $\pi^*$. Two cases:
> > - If $w \pi^* < w \pi_k - \epsilon$, then $\pi_k \succ \pi^*$, so $\pi_k$ will be returned, and also $w \pi_k \ge w \pi_i - \epsilon$ for all $i \in [k]$, so $\pi_k$ is $\epsilon$-optimal over all $k$ policies.
> > - Otherwise, $w \pi^* \ge w \pi_k - \epsilon$, in which case $\pi^*$ is returned, we have $w \pi^* \ge w \pi_i - \epsilon$ for all $i \in [k]$, so $\pi^*$ is $\epsilon$-optimal over all $k$ policies.

---

> > > ### Author Response · Authors · 2023-08-11
> > > **Replying to \epsilon optimality**
> > >
> > > We thank the reviewer for reading our rebuttal and responding promptly. We agree with the reviewer that $O(k)$ is enough to guarantee $\epsilon$ optimality by the induction method mentioned by the reviewer! We will incorporate this into the final version and update the results.

---

### Official Review · Reviewer_wfQV · 2023-07-19

**Soundness:** 3 good
**Presentation:** 4 excellent
**Contribution:** 3 good
**Rating:** 5
**Confidence:** 3

**Summary:**

This paper proposes a method for multi-objective RL considering two user feedback models and thereby the first algorithms with theoretical guarantees for personalized MO decision making via policy comparison.

**Strengths:**

The paper prpoposes an interesting method for finding a personalized optimal policy for MORL problems reflecting users' unknown preferences over the objectives. It is very well written and scienitifically sound, motivates the work well. Method is presented clearly and carefully. The novelty and significance of the method are well presented.

**Weaknesses:**

The paper lacks conclusions and discusssion about limitations completely and thereby its utility is left a bit open.  The paper also emphasizes that it is important to present the obtained trajectories in an easy format to the user, but does not give any examples how this would be achieved.

**Questions:**

You state that the method learns a user's preference vector using minimal number of queries, and C4 is used when the number is small, but discussion of what is small in the number of trajectories or minimal for queries?

Could you give an intuitive example of how are the easily interpretable queries presented to the user.

What are the limitations of the method?

**Limitations:**

Not really addressed.

---

> ### Author Rebuttal · Authors · 2023-08-10
>
> We would like to thank the reviewer for the informative review. We address the reviewer’s questions below.
>
> > The paper lacks conclusions and discussion about limitations.
>
> As mentioned in our general response, we have deferred discussion to Appendix A due to space limitations. For the final version of the paper, we will move the discussion into the main paper.
>
> > Examples on how to present the obtained trajectories in an easy format to the user.
>
> In the Google maps example (lines 16, 56, 300), each trajectory is a route. In the ChatGPT example (line 50), each trajectory is some text generated by a ChatGPT session with a different history. We will make sure to explain this in the final version of the paper.
>
> > What is small in the number of trajectories or minimal for queries or minimal for queries?
>
> By “small” we mean polynomial in $|S|, |A|, H$.
> As mentioned in lines 335-338, our goal is to compress a set of trajectories to the desired size in time $O(poly(H |S| |A|)))$ while the running time of C4 is linear in the size of the input trajectories (stated in Lemma 4), which can be $\Omega(|S|^H)$. Hence, we adopt C4 as a subroutine when the number of trajectories is small. Thus, “small” here refers to $poly(H |S| |A|)$.
>
> Regarding ‘minimal’, a clear lower bound on the query complexity is $\Omega(k+\log(1/\epsilon))$ queries. Our upper bound comes close to it with $O(k\log(k/\epsilon)$ queries.
>
> >Limitations of the method.
>
> We have included some discussions of limitations in Appendix A (which we will move to the main paper).

---

### Author Rebuttal · Authors · 2023-08-10

We would like to thank the reviewers for their valuable and informative reviews! Here we address some common questions.

> Response on abrupt ending of the paper

Due to space limitations, we have deferred the discussion section to Appendix A, where we discuss the results obtained in this work and open problems. We briefly indicated the deferral of the discussion to the appendix immediately after the related work section (line 87) due to space limitations. We will add the discussion section back to the end of the main paper in the final version.

> Response on noisy feedback model to Reviewer 5457 and Reviewer pTmr

Noisy feedback models would be an interesting future direction. If we consider the simplistic noisy model where the feedback is flipped with some fixed probability $\eta< 0.5$, then we can simply repeat each specific query for $1/(0.5-\eta)^2$ times to obtain the true feedback with high probability. Subsequently, the techniques outlined in this paper could be applied. More complex noise models are interesting for future work.

> Response on lack of empirical evidence to Reviewer 5457 and Reviewer BDkG

We would like to address that this work is purely theoretical and our focus is on theoretical analysis.

> Response on running examples to Reviewer wfQV and Reviewer BDkG

We will extend our usage of the Google maps example (lines 16, 56, 300) through the paper to improve the paper's readability. Thank you for your suggestion.

If we misunderstood any of the questions, we would be happy to clarify any further information during the discussion period. Overall, your feedback is valuable, and we appreciate the opportunity to enhance the paper's coherence and readability!

---

### Decision · Program_Chairs · 2023-09-21

**Decision:**

Accept (poster)

**Comment:**

This paper considers a multi-objective RL framework that can model different user preferences over objectives, where these preferences are learned via policy comparisons. The paper proposes algorithms to learn the near-optimal policy for two kinds of feedback and characterizes their query complexity.

The reviewers tend to agree that the paper is well-written, and its contributions merit acceptance. I agree and recommend the acceptance of this paper. Please incorporate the reviewer's comments in the final version of the paper. In particular, addressing the following concerns will help strengthen the current version of the paper:
- Add the discussion in Appendix A to the main paper.
-  Expand on the Google Maps running example, and better justify the proposed feedback model using more examples (response to Rev. 5457, Rev. pTmr)
- Update the leading factor of our computational complexity of Alg 1, update the proof of Lemma 1, and add the discussion about the better approximation of $\epsilon$ (response to Rev. Gnoo)
- Include the reference (Marinescu et al, 2017) in the paper.